# SwiReasoning: Switch-Thinking in Latent and Explicit for Pareto-Superior Reasoning LLMs

**Dachuan Shi[1], Abedelkadir Asi[2], Keying Li[2], Xiangchi Yuan[1], Leyan Pan[1],**
**Wenke Lee[1†], Wen Xiao[2†]**
[1]Georgia Tech    [2]Microsoft    github.com/sdc17/SwiReasoning    swireasoning.github.io

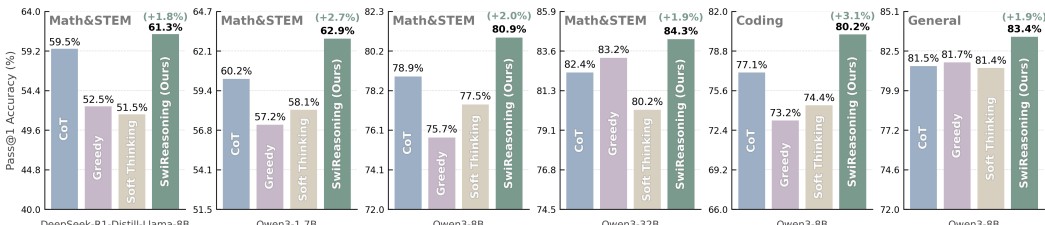

Figure 1: Pass@1 **accuracy** under *unlimited* token budgets. Across the Math&STEM, Coding, and General reasoning benchmarks, SwiReasoning yields average gains of $+2.1\%$, $+3.1\%$, and $+1.9\%$, respectively.

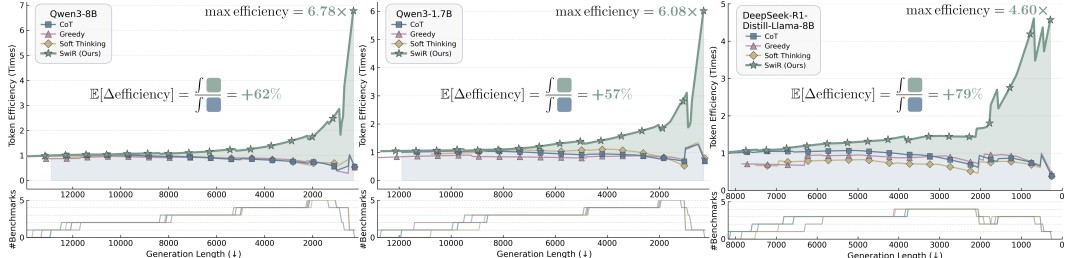

Figure 2: Token **efficiency** (accuracy per token compared to standard CoT), under *limited* token budgets. Across reasoning LLM families and sizes, SwiReasoning brings average efficiency improvements of up to $+79\%$.

## Abstract

Recent work shows that, beyond discrete reasoning through explicit chain-of-thought steps, which are limited by the boundaries of natural languages, large language models (LLMs) can also reason continuously in latent space, allowing richer information per step and thereby improving token efficiency. Despite this promise, latent reasoning still faces two challenges, especially in training-free settings: 1) purely latent reasoning broadens the search distribution by maintaining multiple implicit paths, which diffuses probability mass, introduces noise, and impedes convergence to a single high-confidence solution, thereby hurting accuracy; and 2) overthinking persists even without explicit text, wasting tokens and degrading efficiency. To address these issues, we introduce SwiReasoning, a training-free framework for LLM reasoning which features two key innovations: 1) SwiReasoning dynamically switches between explicit and latent reasoning, guided by block-wise confidence estimated from entropy trends in next-token distributions, to balance exploration and exploitation and promote timely convergence. 2) By limiting the maximum number of thinking-block switches, SwiReasoning curbs overthinking and improves token efficiency across varying problem difficulties. On widely used mathematics, STEM, coding, and general benchmarks, SwiReasoning consistently improves average accuracy by 1.8%–3.1% across reasoning LLMs of different model families and scales. Furthermore, under constrained budgets, SwiReasoning improves average token efficiency by 57%-79%, with larger gains as budgets tighten.

---

†Equal advising.

# 1 INTRODUCTION

Reasoning is one of the central capabilities of large language models (LLMs) (Yang et al., 2025; Qwen Team, 2024; Meta, 2025a;b). It allows models to tackle complex tasks such as mathematics, science, and programming (Guo et al., 2025; OpenAI, 2025b; Jaech et al., 2024; Agarwal et al., 2025; Qwen Team, 2025; Abdin et al., 2025; Abouelenin et al., 2025; Anthropic, 2025; DeepMind, 2024a;b), far beyond simple question answering.

A key limitation of the dominant reasoning approach, explicit chain-of-thought (CoT) (Wolf et al., 2020; Wei et al., 2022; Yao et al., 2023a; Goyal et al., 2024; Pfau et al., 2024), lies in the reliance on discrete tokens during inference. In standard CoT decoding, the model commits to a single token at each step, sampled from the predicted distribution. While effective and ensures readability by verbalizing intermediate steps, this discrete process collapses the full probability distribution into a single trajectory, discarding uncertainty and eliminating many potentially useful reasoning paths.

To overcome this bottleneck, recent work has explored an alternative reasoning technique, latent reasoning (Hao et al., 2024; Zhang et al., 2025; Cheng & Van Durme, 2024; Xu et al., 2025a;b; Tan et al., 2025), where the model operates directly in a continuous hidden space instead of a discrete text space. Latent reasoning offers two key advantages over CoT: 1) higher representational bandwidth per step, since hidden vectors can encode richer information than single tokens (Zhu et al., 2025b); and 2) the ability to preserve multiple reasoning hypotheses implicitly, rather than collapsing them prematurely into one tokenized path (Li et al., 2025b; Chen et al., 2025).

Latent reasoning can be broadly categorized into training-required and training-free approaches. Training-required ones (Hao et al., 2024; Su et al., 2025; Liu et al., 2024; Shen et al., 2025; Tack et al., 2025) demand substantial retraining or fine-tuning (Yue et al., 2025; Li et al., 2025a; Wang et al., 2025a; Zhu et al., 2025a), making it excessively expensive to apply to large reasoning language models. In contrast, training-free approaches like Soft-Thinking (Zhang et al., 2025), which form a probability-weighted mixture of token embeddings as inputs, operate directly at inference time without incurring additional training costs. Our work focuses on the latter category, which is cost-effective and resource-friendly for deployment in large-scale reasoning models.

Although training-free latent reasoning eliminates the need for costly retraining, operating purely in the latent space also presents significant challenges. First, the model is not explicitly trained to perform long-horizon reasoning with latent inputs. As a result of distributional mismatches, when inference relies solely on latent trajectories, the process is less controlled and can easily drift off course (Chen et al., 2025). Instead of collapsing into a single path, the model tends to spread probability mass across many implicit reasoning paths. While this preserves multiple hypotheses, it also introduces persistent noise, slows convergence, and ultimately harms reasoning accuracy (Li et al., 2025b). Second, the absence of explicit tokens does not necessarily ensure efficiency. In latent space, models may still suffer from repetitive or unnecessarily extended internal deliberations and continuation (Zhang et al., 2025), essentially overthinking. This prolongs inference and over-consumes tokens, undermining the very efficiency that latent reasoning is meant to improve.

To address these issues, this paper introduces SWIREASONING (abbreviated as SWIR) as a training-free framework for LLM reasoning that alternates between explicit and latent thinking, based on block-wise confidence inferred from entropy trends of next-token distributions, and suppresses overthinking by bounding the number of switches. More specifically, the framework first tracks a reference entropy within each thinking block to reflect block-wise confidence. Rising confidence triggers an explicit switch to consolidate progress along a single path, while sustained uncertainty triggers a latent switch to re-explore in continuous space. Second, a switch count controller caps the number of thinking block transitions and provides early-answer checkpoints, curbing unnecessary latent loops and improving token efficiency across difficulties.

The proposed framework also benefits from reintroducing diversity by sampling in an explicit thinking block when compared to pure latent thinking. Even though motivated differently, SWIREASONING resonates with the concurrent observation of Wu et al. (2025b) that introducing stochasticity benefits latent reasoning, but we achieve this via a distinct mode switch mechanism rather than injecting distributions with randomness.

Our contributions are summarized as follows:

- We propose SWIREASONING, a training-free reasoning framework that dynamically alternates between explicit and latent thinking based on confidence signals, thereby exploiting the expressivity of latent thinking without sacrificing the stability of explicit thinking.

- We introduce a switch count control mechanism that caps the number of transitions, enabling early answering based on partial reasoning trajectories at switch boundaries. This effectively suppresses overthinking and improves token efficiency under limited budgets.

- We extensively validate the effectiveness of SWIREASONING on mathematics, STEM, coding, and general reasoning domains across multiple benchmarks, model families, and sizes, demonstrating consistent gains in both accuracy and token efficiency over training-free baselines.

## 2 RELATED WORK

**Explicit LLM Reasoning.** Reasoning via explicit intermediate text has been extensively studied. Chain-of-thought (CoT) prompting elicits stepwise rationales that improve reasoning accuracy by decomposing problems into natural-language sub-steps (Kojima et al., 2022; Wei et al., 2022). Subsequent work increases robustness by aggregating multiple CoT trajectories through self-consistency (Wang et al., 2022). Search- and tool-augmented variants further expand the exploration space, such as Tree-of-thought that branches over partial rationales (Yao et al., 2023a) and interleaving reasoning and actions with external tools and environments (Yao et al., 2023b). Least-to-most prompting progressively solves subproblems to reduce reasoning load and mitigate error accumulation (Zhou et al., 2022). These approaches operate purely in the discrete token space and therefore commit to a single token at each step. While readable, the discretizations in explicit reasoning discard alternative hypotheses early, and restrict the information bandwidth per step (Zhu et al., 2025b).

**Latent LLM Reasoning.** Latent reasoning operates in the continuous representation space rather than discrete natural language space used by explicit reasoning. Prior work can be broadly grouped into two categories. 1) Training-required approaches modify pretraining (Tack et al., 2025; Zeng et al., 2025) or fine-tuning objectives (Tan et al., 2025; Wang et al., 2025a;b; Jiang et al., 2025; Wu et al., 2025a; Yue et al., 2025; Li et al., 2025a; Shen et al., 2025; Xu et al., 2025a) to supervise hidden-state trajectories or to endow models with latent-planning skills. 2) Training-free approaches (Zhang et al., 2025; Wu et al., 2025b) intervene only at inference time by manipulating hidden representations or probability distributions without updating model weights. Our work belongs to the training-free category but differs from prior single-mode methods. Instead of remaining purely latent, SWIREASONING dynamically switches between latent and explicit reasoning based on entropy-trend confidence, and further regulates the number of switches through a count controller to suppress overthinking and improve efficiency.

## 3 METHODOLOGY

### 3.1 SWIREASONING OVERVIEW

As shown in Fig. 3, SWIREASONING is a training-free framework that dynamically alternates between explicit and latent reasoning. The number of switches is regulated to suppress overthinking and improve token efficiency. Sec. 3.2 presents the preliminaries of explicit and latent reasoning, Sec. 3.3 details the design of the dynamic switch, and Sec. 3.4 discusses the switch count control mechanism. Implementation details are provided in Appendix B.1.

### 3.2 PRELIMINARY: EXPLICIT AND TRAINING-FREE LATENT THINKING

**Explicit Thinking.** Let $V$ be a vocabulary and $p_\theta(x_t \mid x_{<t})$ an LLM over $V$ with parameters $\theta$. Given a question $q$, the model produces a reasoning trace $r_{1:T} \in V^T$ followed by a final answer $a_{1:U} \in V^U$. We write the concatenated sequence as

$$x_{1:(|q|+T+U)} = [q, r_{1:T}, a_{1:U}],$$

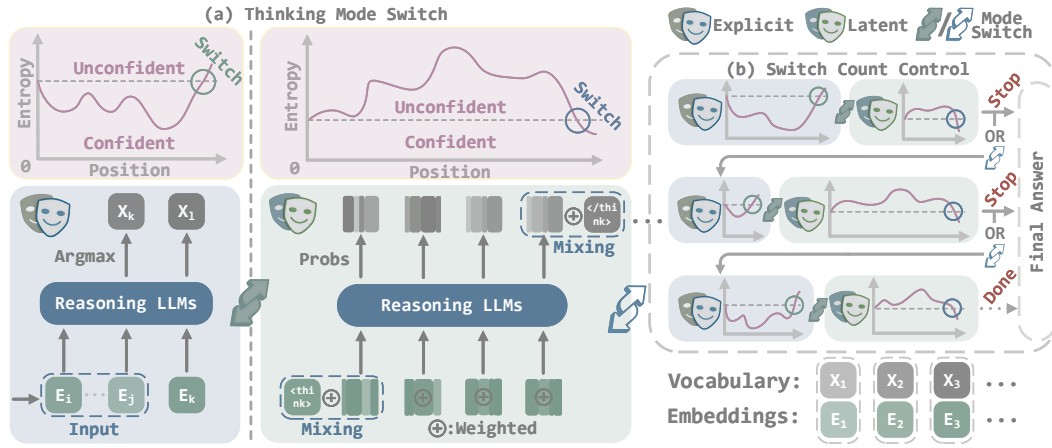

Figure 3: SWIREASONING framework. (a) Dynamic mode switching alternates between explicit and latent thinking based on block-wise confidence estimated from entropy trends. (b) A switch count control mechanism limits the maximum number of thinking-block transitions, suppressing overthinking before the final answer.

At inference, decoding proceeds by repeatedly choosing a token $x_t$ from the predictive distribution $p_\theta(\cdot \mid x_{<t})$ according to a policy $\pi_t(\cdot)$, e.g.,

$$x_t \sim \pi_t(\cdot) \text{ with } \pi_t = \begin{cases} \text{Greedy: } \arg\max_{v \in V} p_\theta(v \mid x_{<t}), \\ \text{Sampling: Top-}k/\text{Top-}p \text{ with temperature } \tau. \end{cases}$$

The reasoning phase stops when a termination condition is met, e.g., generating $\langle$/think$\rangle$, after which the answer tokens $a_{1:U}$ are decoded in the same manner. While explicit reasoning improves reliability by externalizing intermediate steps, its hard policy $\pi_t(\cdot)$ collapses the full distribution to a single discrete decision at each step, i.e., discards information in $p_\theta(\cdot \mid x_{<t})$ beyond the chosen token.

**Training-Free Latent Thinking.** It replaces the hard policy $\pi_t(\cdot)$ by a continuous surrogate that preserves distributional information. Let $E \in \mathbb{R}^{|V| \times d}$ denote the token embedding matrix with rows $e^{(v)} \in \mathbb{R}^d$. At step $t$, the model yields logits $\ell_t \in \mathbb{R}^{|V|}$ and $p_t = \text{softmax}(\ell_t)$. Given the next-token distribution $p_t := p_\theta(\cdot \mid x_{<t}) \in \Delta^{|V|-1}$, it forms a soft embedding

$$\tilde{e}_t = \sum_{v \in V} p_t[v]\, e^{(v)} \in \mathbb{R}^d, \tag{1}$$

and feeds $\tilde{e}_t$ back to the model as the next input representation, rather than committing to an explicit token by $\pi_t(\cdot)$. Upon the thinking phase being complete, the policy reverts to $\pi_t(\cdot)$ for answer generation. The convexity of Eq. 1 ensures $\tilde{e}_t$ lies in the embedding hull of $E$, retaining all first-order uncertainty in $p_t$, which reduces information discards and increases robustness to local noise.

### 3.3 DYNAMIC SWITCH BETWEEN EXPLICIT AND LATENT THINKING

Remaining in a single mode throughout reasoning is inherently suboptimal: explicit thinking provides readability but may discard useful information beyond chosen tokens, while latent thinking preserves richer signals but can drift into noise and reduce accuracy. Our key insight is that *reasoning should switch modes based on confidence*. Latent reasoning enables exploration across multiple potential continuations when confidence is low, and explicit reasoning encourages convergence when confidence is high, striking a balance that supports broad exploration while maintaining accuracy.

**Mode Switch Criterion.** We refer to the reasoning content between two consecutive switches as a thinking block and estimate its confidence by entropy $H_t = -\sum_v p_t[v] \log p_t[v]$. Let $\bar{H}$ denote the reference entropy of the current block, which is initialized at the first step of the block and refreshed when a mode switch happens. We use a criterion that converts local entropy trends into decisions:

$$\text{Latent} \rightarrow \text{Explicit}: \quad (H_t < \bar{H}) \quad \text{(confidence rises)}, \tag{2}$$

$$\text{Explicit} \rightarrow \text{Latent}: \quad (H_t > \bar{H}) \quad \text{(confidence drops)}, \tag{3}$$

**Switch Window Size.** To avoid oscillations, we impose dwell windows upon the mode switch criterion. Formally, with mode variable $m_t \in \{\text{Explicit}, \text{Latent}\}$ and dwell step counter $\Delta t$, we have

$$m_{t+1} = \begin{cases} \text{Explicit}, & m_t = \text{Latent} \wedge (H_t < \bar{H}) \wedge (\Delta t \geq W_{\text{L}\rightarrow\text{E}}), \\ \text{Latent}, & m_t = \text{Explicit} \wedge (H_t > \bar{H}) \wedge (\Delta t \geq W_{\text{E}\rightarrow\text{L}}), \\ m_t, & \text{otherwise.} \end{cases}$$

We reset $\bar{H} \leftarrow H_t$, $\Delta t \leftarrow 0$ upon any switch, *i.e.*, $m_{t+1} \neq m_t$. Otherwise, we update $\Delta t \leftarrow \Delta t + 1$. In practice, $W_{\text{L}\rightarrow\text{E}} = 0$ while $W_{\text{E}\rightarrow\text{L}}$ is positive, *i.e.*, a Latent→Explicit switch may occur immediately when $H_t$ dips, whereas an Explicit→Latent switch requires staying for at least $W_{\text{E}\rightarrow\text{L}}$ steps.

The key intuition behind the asymmetric design is that *two modes play different roles in reasoning*. Latent reasoning is inherently divergent, allowing for rich exploration. However, prolonging the latent phase after confidence has recovered is counterproductive. It increases the risks of introducing spurious signals that may mislead the model. Therefore, once confidence rises, an immediate switch back to explicit reasoning is necessary to consolidate progress onto a single coherent trajectory.

In contrast, explicit reasoning is convergent, gradually unfolding a chain-of-thought where each token incrementally extends the current logical path. If the model were allowed to switch back to latent reasoning at the first sign of an entropy fluctuation, spurious short-term uncertainty could trigger oscillations. The dwell window $W_{E\rightarrow L}$ ensures that explicit reasoning is given sufficient opportunity to stabilize and accumulate meaningful structure.

**Thinking-Related Signal Mixing.** To better align mode switches with the LLMs' learned reasoning patterns, we blend the embeddings of thinking-related signal tokens, *e.g.*, `<think>` and `</think>`, when a switch occurs. Let $e_{\langle\text{think}\rangle}$ and $e_{\langle/\text{think}\rangle}$ denote their embeddings. At the entrance to a latent thinking block, we bias the first latent step $t^\star$ toward "begin thinking" by

$$\tilde{e}_{t^\star} \leftarrow \alpha_{t^\star} \cdot \tilde{e}_{t^\star} + (1 - \alpha_{t^\star}) \cdot e_{\langle\text{think}\rangle}, \qquad \alpha_{t^\star} \in [0, 1], \tag{4}$$

and at the exit to an explicit thinking block, we bias the first explicit step $t^\dagger$ toward "end thinking"

$$\tilde{e}_{t^\dagger} \leftarrow \beta_{t^\dagger} \cdot \tilde{e}_{t^\dagger} + (1 - \beta_{t^\dagger}) \cdot e_{\langle/\text{think}\rangle}, \qquad \beta_{t^\dagger} \in [0, 1], \tag{5}$$

which encourages the model to close the latent phase and move on to answer production. In practice, we schedule $\alpha_t = \alpha_0 + (1 - \alpha_0)\frac{t}{T_{\max}}$ and $\beta_t = \beta_0 + (1 - \beta_0)\frac{t}{T_{\max}}$, where $T_{\max}$ is a predefined maximum generation length, and apply Eq. 4 or Eq. 5 only at the steps when switches occur.

### 3.4 Overthinking Suppression by Switch Count Control

Even with confidence-aware switching, reasoning LLMs may still overthink. Therefore, we place a bound on the total number of Latent→Explicit switches. Our key insight is that each switch naturally marks *the end of a thinking block where partial reasoning trajectories have been consolidated*, which may already contain sufficient evidence for arriving at a reasonable solution. Under limited budgets, generating answers at these natural checkpoints can make use of partial reasoning trajectories, offering a chance to obtain correct predictions earlier without consuming additional tokens.

**Counter and Triggers.** Let $C_t$ count completed Latent→Explicit switches up to step $t$. Given a user-specified budget $C_{\max}$, we define two triggers:

- **Convergence trigger** (at $\frac{1}{2}C_{\max} \leq C_t \leq C_{\max}$ on Latent→Explicit transitions): force the next token to be $\langle/\text{think}\rangle$. The convergence trigger is to encourage rather than enforce the end of the thinking process and the start of converging to an answer based on partial reasoning trajectories.

- **Termination trigger** (at $C_t > C_{\max}$ on a subsequent Latent→Explicit transition): inject a concise answer prefix $s_{\text{final}}$, "$\langle/\text{think}\rangle$\n\n The final answer is", then allow at most $B$ additional tokens for the final answer. The termination trigger is to enforce an immediate answer generation.

Triggers are implemented as short injection queues that overwrite future-generated tokens. Formally, let $\mathcal{Q}_t$ be the per-sample injection queue. When a convergence or termination trigger fires, we set $\mathcal{Q}_t \leftarrow [\text{ID}(\langle/\text{think}\rangle)]$ or $[\text{ID}(s_{\text{final}})]$. At the next step, if $\mathcal{Q}_t \neq \emptyset$, we deterministically set $x_t \leftarrow \mathcal{Q}_t.\text{pop}()$. For the termination trigger, we also start a budget counter $b_t = B$ and decrement it each step after the termination trigger fires. Decoding will be terminated once $b_t = 0$.

# 4 EXPERIMENTS

## 4.1 EXPERIMENTAL SETTINGS

**Models.** We evaluate SWIREASONING on four recent reasoning LLMs: DeepSeek-R1-Distill-Llama-8B (Guo et al., 2025), Qwen3-1.7B (Yang et al., 2025), Qwen3-8B (Yang et al., 2025), and Qwen3-32B (Yang et al., 2025). This selection helps us validate the effectiveness of SWIREASONING across different model families, model scales, and training paradigms.

**Domains and Benchmarks.** We evaluate SWIREASONING on 11 benchmarks spanning four domains: mathematical reasoning (GSM8K (Cobbe et al., 2021), Math500 (Hendrycks et al., 2021), AIME 2024 (HuggingFaceH4, 2024), AIME 2025 (Yentinglin, 2025)); STEM reasoning (GPQA Diamond (Rein et al., 2024)); coding reasoning (HumanEval (Chen et al., 2021), LeetCode-Contest (Guo et al., 2024), MBPP (Austin et al., 2021), LiveCodeBench (Jain et al., 2024)); and general reasoning (2WikiMultihopQA (Ho et al., 2020), CommonsenseQA (Talmor et al., 2019)).

**Baselines.** We compare SWIREASONING that dynamically switches between thinking modes against three baselines with a single thinking mode, including 1) explicit thinking alone: standard CoT reasoning with sampling, standard CoT reasoning with greedy decoding, and 2) training-free latent thinking alone: Soft Thinking (Zhang et al., 2025).

**Metrics.** We use the Pass@1 metric to evaluate reasoning accuracy, and token efficiency $E$ to assess the level of reasoning efficiency. Let $\text{Acc}_m(\ell) \in [0, 1]$ denote the accuracy of method $m$ when using $\ell$ generated tokens. Its plain token efficiency is the accuracy gained per token,

$$PE_m(\ell) \ = \ \frac{\text{Acc}_m(\ell)}{\ell}.$$

To express efficiency in units relative to the standard CoT, we normalize it by the CoT's plain token efficiency when the highest accuracy is achieved. Specifically, if CoT achieves its highest accuracy $\text{Acc}^\star_{\text{CoT}}$ using $\ell^\star_{\text{CoT}}$ tokens, denote $PE^\star_{\text{CoT}} = \frac{\text{Acc}^\star_{\text{CoT}}}{\ell^\star_{\text{CoT}}}$. The token efficiency of $m$ is defined as

$$\text{E}_m(\ell) \ = \ \frac{PE_m(\ell)}{PE^\star_{\text{CoT}}} \ = \ \frac{\text{Acc}_m(\ell)/\ell}{\text{Acc}^\star_{\text{CoT}}/\ell^\star_{\text{CoT}}}.$$

And the average efficiency gain of method $m$ over CoT is

$$\mathbb{E}[\Delta\text{E}_m] \ = \ \frac{\int (\text{E}_m(\ell) - \text{E}_{\text{CoT}}(\ell))\, d\ell}{\int \text{E}_{\text{CoT}}(\ell)\, d\ell}.$$

## 4.2 REASONING ACCURACY UNDER UNLIMITED TOKEN BUDGETS

We first evaluate SWIREASONING in the setting where token budgets are set large enough to ensure that each method is allowed to conduct sufficient thinking (refer to Appendix B.2 for detailed settings). Fig. 1 and Tab. 1 report the highest attainable accuracies across mathematics (GSM8K, MATH500, AIME24, AIME25) and STEM (GPQA Diamond) benchmarks under this setting. Across different model families of varying sizes, SWIREASONING consistently achieves higher Pass@1 accuracy than CoT with sampling, CoT with greedy decoding, and Soft Thinking.

Our observation is that improvements are most pronounced on the more challenging benchmarks. For instance, on AIME24/AIME25, which require deep deductive reasoning and are widely regarded as more difficult, our method yields absolute gains of 3.34%/2.50% on Qwen3-8B, and 5.00%/5.00% on Qwen3-1.7B. These margins substantially exceed those observed on GSM8K or MATH500 with lower difficulty, suggesting that the proposed switching mechanism is particularly beneficial when problems involve long reasoning chains or higher uncertainty from the LLM's perspective. Overall, the accuracy results under unlimited token budgets highlight the strength of SWIREASONING in better addressing reasoning tasks compared to single-mode approaches.

Table 1: Comparison of SWIREASONING and CoT with sampling, CoT with greedy decoding, and Soft Thinking on mathematics and STEM benchmarks. SWIREASONING improves accuracy by $+2.17\%$ on average.

| Method | GSM8K | MATH 500 | GPQA Diamond | AIME 2024 | AIME 2025 | Average | |
|---|---|---|---|---|---|---|---|
| *Qwen3-8B* (Yang et al., 2025) | | | | | | | |
| CoT | 95.60 | 96.00 | 59.60 | 75.83 | 67.50 | 78.91 | +0.00 |
| CoT (Greedy) | 95.68 | 96.40 | 56.57 | 70.00 | 60.00 | 75.73 | −3.18 |
| Soft Thinking | 95.38 | 96.00 | 59.60 | 67.92 | 68.33 | 77.45 | −1.46 |
| *SwiR (Ours)* | **96.06** +0.46 | **98.40** +2.40 | **61.11** +1.51 | **79.17** +3.34 | **70.00** +2.50 | **80.94** +2.03 |
| *Qwen3-1.7B* (Yang et al., 2025) | | | | | | | |
| CoT | 90.44 | 92.00 | 39.39 | 45.83 | 33.33 | 60.20 | +0.00 |
| CoT (Greedy) | 89.61 | 91.00 | 31.82 | 40.00 | 33.33 | 57.15 | −3.05 |
| Soft Thinking | 90.30 | 90.60 | 34.34 | 38.75 | 36.67 | 58.13 | −2.07 |
| *SwiR (Ours)* | **90.83** +0.39 | **93.00** +1.00 | **41.41** +2.02 | **50.83** +5.00 | **38.33** +5.00 | **62.88** +2.68 |
| *DeepSeek-R1-Distill-Llama-8B* (Guo et al., 2025) | | | | | | | |
| CoT | 89.46 | 91.40 | 46.46 | 43.75 | 26.25 | 59.46 | +0.00 |
| CoT (Greedy) | 85.82 | 84.80 | 31.81 | 30.00 | 30.00 | 52.49 | −6.97 |
| Soft Thinking | 85.90 | 83.80 | 33.33 | 34.17 | 20.42 | 51.52 | −7.94 |
| *SwiR (Ours)* | **90.07** +0.61 | **92.00** +0.60 | **47.98** +1.52 | **45.00** +1.25 | **31.25** +5.00 | **61.26** +1.80 |

## 4.3 TOKEN EFFICIENCY UNDER LIMITED TOKEN BUDGETS

Across models and benchmarks, SWIREASONING consistently attains improved Pareto frontiers. As shown in Fig. 2, the peak efficiency gains range between $4.6\times$ and $6.8\times$ over CoT depending on the model size. These improvements are not confined to a single budget: the area-under-curve (AUC) advantage persists across a broad range of small to moderate budgets.

One observation from Fig. 4 is that the relatively large average efficiency gains occur on GSM8K, MATH500, and GPQA Diamond across three models (up to $213\%$ AUC improvements in the per-benchmark panels). These tasks contain many instances with lower difficulty, which benefit most from our overthinking suppression design to obtain the correct answer with partial reasoning trajectories. In contrast, on AIME24/25, the efficiency gaps are smaller, while the accuracy gains with unlimited budgets are larger. This asymmetry is expected: the harder the problem is, the more difficult it is to predict a correct answer with unfinished reasoning trajectories. Overall, token efficiency results under limited budgets substantiate the advantage of SWIREASONING in gaining accuracy more efficiently as budgets tighten compared to baseline methods.

## 4.4 EVALUATION WITH PASS@K ACCURACY

In addition to Pass@1 accuracy, we also measure Pass@k accuracy, where $k \in [1, 64]$ on Qwen3-8B. Fig. 5 shows that SWIREASONING reaches its maximal accuracy with significantly smaller $k$ than baselines. Define $k^\star$ as the smallest $k$ achieving the method's peak. On AIME24, we observe $k^\star = 13$ for SWIREASONING versus 46 for CoT (about $72\%$ fewer samples), and on AIME25, $k^\star = 16$ versus 22 (about $27\%$ fewer samples). In addition to the faster growth of the curve than CoT, SWIREASONING also exhibits 1) a steeper initial slope at small $k$ (higher "per-sample yield"), and 2) a higher eventual ceiling than Soft Thinking and greedy CoT, indicating better correctness and diversity simultaneously. Overall, Pass@k accuracy results indicate that SWIREASONING is particularly attractive for budgeted evaluation settings where $k$ cannot be large.

## 4.5 ABLATION STUDIES

**Switch Window Size.** SWIREASONING uses dwell windows (Sec. 3.3) to enforce the model stays in a thinking block for at least $W$ steps before switching to the other thinking mode. We conduct ablation studies on Qwen3-1.7B with a representative setting consisting of $W_{\mathrm{E}\to\mathrm{L}} \in \{64, 128, 256, 512, 1024\}$ and report Pass@1 accuracy on five benchmarks. Results in Tab. 3 demonstrate that an intermediate window size of 512 consistently produces the best results.

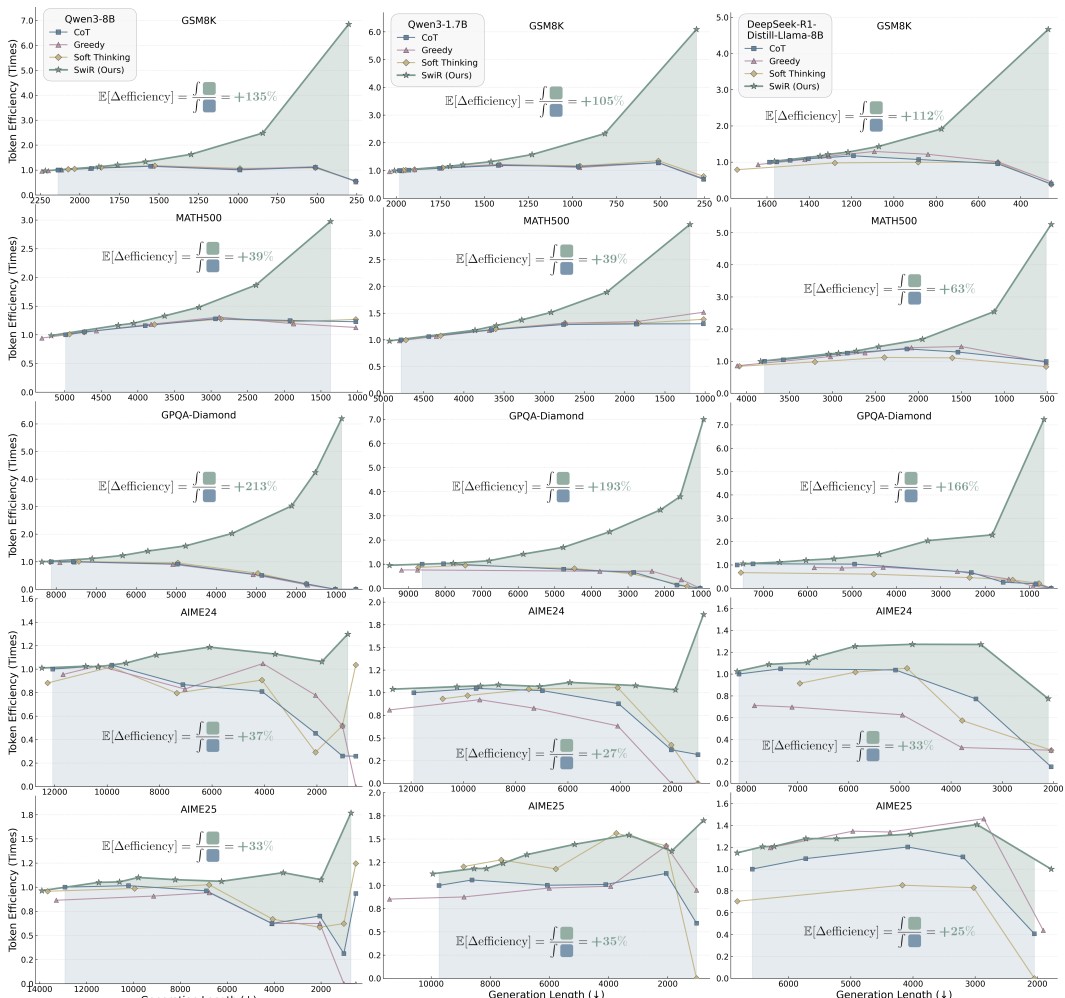

Figure 4: Token efficiency comparisons. SWIREASONING achieves the highest token efficiency throughout all token budgets in 13 out of 15 evaluations, with an efficiency improvement of $+84\%$ over CoT on average.

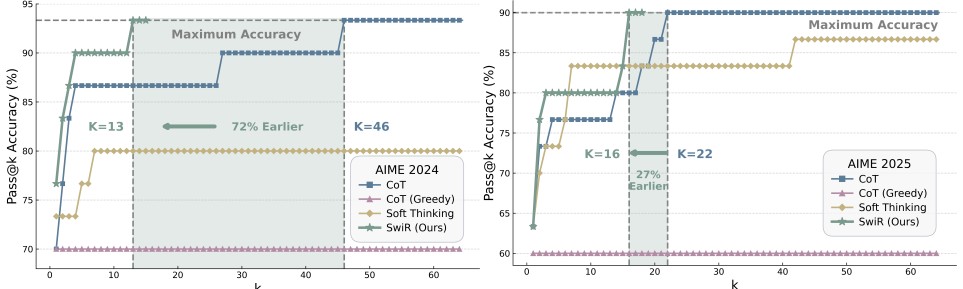

Figure 5: Pass@k accuracy ($k \in [1, 64]$) evaluation with Qwen3-8B on AIME 2024 and 2025 benchmarks. SWIREASONING achieves maximum reasoning accuracies $+50\%$ earlier compared to CoT on average.

When window sizes are too small, LLMs may jump back to latent mode prematurely, before explicit reasoning has consolidated a coherent trajectory. This increases exposure to noisy signals and harms final accuracy, especially on difficult tasks such as AIME24 and AIME25. When window sizes are too large, LLMs become sluggish to reenter latent exploration as confidence declines. A promising improvement direction is to make $W$ adaptive to the model's real-time density of effective reasoning.

**Thinking-Related Signal Mixing.** SWIREASONING uses $\alpha_0, \beta_0 \in [0, 1]$ as the initial ratios for mixing thinking-related signals at switching instants (Sec. 3.3). We sweep $\alpha_0$ and $\beta_0$ independently and report Pass@1 accuracies in Tab. 2.

Table 2: Ablations on $\alpha_0$ and $\beta_0$ for signal mixing. **Greener** indicates better performance within each column.

| $\alpha_0$ | GSM8K | MATH 500 | GPQA Diamond | AIME 2024 | AIME 2025 | Average | $\beta_0$ | GSM8K | MATH 500 | GPQA Diamond | AIME 2024 | AIME 2025 | Average |
|---|---|---|---|---|---|---|---|---|---|---|---|---|---|
| 0.0 | 89.23% | 89.80% | 35.86% | 46.67% | 35.00% | 59.31% | 0.0 | 81.50% | 67.20% | 28.79% | 8.33% | 9.17% | 39.00% |
| 0.1 | 89.84% | 91.00% | 36.36% | 46.25% | 36.25% | 59.94% | 0.1 | 81.88% | 70.20% | 31.82% | 11.67% | 8.75% | 40.86% |
| 0.2 | 90.37% | 91.60% | 34.85% | 46.25% | 37.50% | 60.11% | 0.2 | 82.11% | 70.60% | 28.28% | 14.17% | 9.17% | 40.87% |
| 0.3 | 90.45% | 91.60% | 38.38% | 47.08% | **38.33%** | 61.17% | 0.3 | 90.67% | 92.00% | 37.37% | 45.42% | 37.92% | 60.68% |
| 0.4 | 89.61% | 92.80% | 40.91% | 48.33% | 32.50% | 60.83% | 0.4 | **90.98%** | 91.40% | 37.88% | 47.92% | 36.67% | 60.97% |
| 0.5 | 90.45% | **93.00%** | 34.34% | **50.83%** | 36.25% | 60.97% | 0.5 | 90.37% | 91.20% | **42.42%** | 47.92% | 35.83% | 61.55% |
| 0.6 | **90.83%** | 92.00% | 39.39% | 44.58% | 37.92% | 60.94% | 0.6 | 90.59% | 90.40% | **42.42%** | 42.50% | 36.67% | 60.52% |
| 0.7 | 90.06% | 91.60% | 37.37% | 45.00% | 37.08% | 60.22% | 0.7 ✓ | 90.83% | **93.00%** | 41.41% | **50.83%** | 38.33% | **62.88%** |
| 0.8 | 90.60% | 92.00% | 37.37% | 48.33% | 35.42% | 60.74% | 0.8 | 89.99% | 92.20% | 39.39% | 49.17% | 35.83% | 61.32% |
| 0.9 | 90.37% | 90.80% | 39.39% | 50.42% | 35.83% | 61.36% | 0.9 | 90.22% | 92.20% | 40.91% | 48.75% | 32.50% | 60.52% |
| 1.0 ✓ | 90.14% | 90.60% | **41.41%** | 49.17% | 37.92% | **61.85%** | 1.0 | 90.44% | 91.00% | 33.33% | 46.67% | **38.75%** | 60.04% |

For the exit bias $\beta_0$, a very small $\beta_0$ implies excessive interference with when to conclude the thinking process and severely degrades accuracy (*e.g.*, AIME24 drops to 8.33% at $\beta_0=0.0$). Performance rises sharply and peaks near $\beta_0=0.7$, which achieves the best average 62.88% and is either the best or the second-best on most datasets. A promising improvement direction is to make $\beta_0$ difficulty-aware, so that it will be automatically adjusted based on problem difficulty.

The situation for the entrance bias $\alpha_0$ is different. We observe a broad performance plateau for $\alpha_0 \in [0.4, 0.9]$, with the highest average at $\alpha_0=1.0$ (61.85%), however, only marginally higher than other values like $\alpha_0=0.9$ (61.36%). Task-wise, problems with different levels of difficulty tend to have various preferences over $\alpha_0$. We expose $\alpha_0$ to users for adjustment based on task difficulty. The more detailed hyperparameters we adopted for the experiments are provided in Appendix B.3.

**Maximum Switch Count.** We suppress overthinking by bounding the number of mode switches with a budget $C_{\max}$ (Sec. 3.4), and reducing $C_{\max}$ leads to earlier convergence. In Fig. 2 and Fig. 4, moving rightward on the $x$–axis corresponds to smaller token budgets, *i.e.*, smaller $C_{\max}$. We collect data points in these figures by incrementing the value of $C_{\max}$ from 1 until further increases in $C_{\max}$ no longer alter generation results in most cases, *i.e.*, maximum accuracy is reached at saturation. Detailed data is provided in Appendix C.8.

Table 3: Ablation on switch window size. **Greener** indicates better performance within each column.

| Window Size | GSM8K | MATH 500 | GPQA Diamond | AIME 2024 | AIME 2025 | Average |
|---|---|---|---|---|---|---|
| 64 | 89.69% | 92.60% | 40.91% | 47.92% | 34.17% | 61.06% |
| 128 | 90.45% | 91.00% | 38.89% | 48.33% | 36.25% | 60.98% |
| 256 | 89.76% | 90.80% | 39.90% | 49.58% | 36.25% | 61.26% |
| 512 ✓ | **90.83%** | **93.00%** | **41.41%** | **50.83%** | **38.33%** | **62.88%** |
| 1024 | **90.83%** | 91.20% | 40.40% | 49.58% | 36.67% | 61.74% |

As analyzed in Sec. 4.3, decreasing $C_{\max}$ yields a significant improvement in token efficiency, which confirms the intended behavior of the switch count control design: it curbs prolonged latent exploration and commits to an answer path early, thereby mitigating overthinking. With switch count control, a small number of confidence-aware blocks usually suffices for easy-to-moderate problems, while difficult instances benefit more from allowing a few more switches before the final answer.

More ablation studies are provided in Appendix C.5–C.7.

## 4.6 EXPERIMENTAL RESULTS ON LARGER MODELS

Table 4: Comparison of SWIREASONING and CoT with sampling, CoT with greedy decoding, and Soft Thinking on Qwen3-32B. SWIREASONING improves accuracy by $+1.92\%$ on average.

| Method | GSM8K | MATH 500 | GPQA Diamond | AIME 2024 | AIME 2025 | Average | |
|---|---|---|---|---|---|---|---|
| CoT | 95.83 | 97.40 | 66.16 | 80.42 | 72.08 | 82.38 | +0.00 |
| CoT (Greedy) | 95.91 | 97.20 | 69.70 | 80.00 | 73.33 | 83.23 | +0.85 |
| Soft Thinking | 95.75 | 97.40 | 67.17 | 74.58 | 66.25 | 80.23 | −2.15 |
| *SwiR (Ours)* | **96.21** +0.38 | **98.40** +1.00 | **70.20** +4.04 | **82.92** +2.50 | **73.75** +1.67 | **84.30** +1.92 |

In addition to the 1.7B and 8B settings, we evaluate SWIREASONING on the 32B scale using Qwen3-32B. We report pass@1 accuracy on GSM8K, MATH500, GPQA Diamond, AIME 2024, and AIME 2025. Baseline hyperparameters follow the recommendations from their original papers, and all methods use the same configuration as in the main paper. Table 4 shows that under no token budget constraint, SWIREASONING improves accuracy by $+1.92\%$ on average over standard CoT. The gains are most notable on more difficult benchmarks such as GPQA Diamond ($+4.04\%$). These results indicate that SWIREASONING scales to larger models and achieves consistent accuracy gains.

## 4.7 EXPERIMENTAL RESULTS ON BROADER DOMAINS

Table 5: Comparison of SWIREASONING and baselines on coding, multi-hop QA, and commonsense reasoning tasks. SWIREASONING improves accuracy by $+2.70\%$ on average.

| Method | HumanEval | LeetCode-Contest | | | MBPP |
| --- | --- | --- | --- | --- | --- |
| | | *Easy-level* | *Medium-level* | *Hard-level* | |
| CoT | 92.68 | 57.78 | 68.13 | 43.18 | 94.16 |
| CoT (Greedy) | 93.90 | 64.44 | 58.24 | 47.73 | 91.44 |
| Soft Thinking | 92.07 | 55.56 | 61.54 | 38.64 | 94.16 |
| *SwiR (Ours)* | **95.73** ▮+3.05 | **64.44** ▮+6.66 | **69.23** ▮+1.10 | **61.36** ▮+18.18 | **95.33** ▮+1.17 |

| Method | LiveCode-Bench | 2WikiMul-tihopQA | Common-senseQA | Average | |
| --- | --- | --- | --- | --- | --- |
| CoT | 62.01 | 79.00 | 83.95 | 78.54 | +0.00 |
| CoT (Greedy) | 50.18 | 79.50 | 83.95 | 76.03 | −2.51 |
| Soft Thinking | 56.99 | 79.00 | 83.70 | 76.73 | −1.81 |
| *SwiR (Ours)* | **63.44** ▮+1.43 | **81.50** ▮+2.50 | **85.34** ▮+1.39 | **81.24** ▮+2.70 | |

In addition to math and STEM domains, we further evaluate SWIREASONING on coding, multi-hop QA, and commonsense reasoning. We use Qwen3-8B and report pass@1 accuracy on HumanEval (Chen et al., 2021), LeetCode-Contest (Guo et al., 2024), MBPP (Austin et al., 2021), and LiveCodeBench (Jain et al., 2024) for coding, 2WikiMultihopQA (Ho et al., 2020) set from LongBench (Bai et al., 2024) for multi-hop QA, and CommonsenseQA (Talmor et al., 2019) for commonsense reasoning. As shown in Tab. 5, under no token budget constraint, SWIREASONING improves accuracy by $+2.70\%$ on average over standard CoT (the reported average is the simple mean of accuracy over the full LeetCode-Contest and accuracies over five other benchmarks).

On coding tasks, the largest gains ($+18.18\%$) are observed on the hard-level subset, indicating that SWIREASONING is most helpful for problems that require stronger reasoning capabilities. On multi-hop QA tasks, which require retrieving and connecting multiple disparate facts, SWIREASONING outperforms CoT by $+2.50\%$. This suggests that the exploration capability of latent reasoning is effective in navigating complex reasoning paths. On commonsense reasoning tasks, SWIREASONING surpasses CoT by $+1.39\%$. This demonstrates the robustness of SWIREASONING in general knowledge scenarios. Overall, these results confirm that SWIREASONING generalizes to broader domains with consistent accuracy gains.

## 5 CONCLUSION

This paper presents SWIREASONING, a training-free inference framework that integrates explicit chain-of-thought thinking with latent thinking through an entropy trends–based controller. The framework is conceptually straightforward but empirically effective: when block-wise uncertainty decreases, we collapse to a single explicit path to consolidate progress. When uncertainty rises and has persisted for a minimal dwell window, we expand into latent space to explore more alternatives. Complementing this mode switch, a switch count controller caps the number of transitions, thereby curbing overthinking while preserving prediction quality. Together, these two mechanisms yield consistently improved Pareto frontiers for reasoning LLMs, effectively enhancing both maximum accuracy under unlimited budgets and token efficiency under limited budgets. Looking ahead, integrating SWIREASONING with reinforcement learning–based training may unlock even stronger reasoning capabilities.

ACKNOWLEDGMENTS

The authors would like to thank Microsoft and Georgia Tech for their support. This material is based upon work supported by the National Science Foundation under grant no. 2229876 and is supported in part by funds provided by the National Science Foundation, by the Department of Homeland Security, and by IBM. Any opinions, findings, and conclusions or recommendations expressed in this material are those of the author(s) and do not necessarily reflect the views of the National Science Foundation or its federal agency and industry partners.

## 6 ETHICS STATEMENT

This work focuses on enhancing the reasoning accuracy and token efficiency of LLMs, which does not raise safety concerns. This work involves no collection of sensitive data. All evaluations are conducted using publicly available models and benchmarks under their original licenses.

## 7 REPRODUCIBILITY STATEMENT

We provide implementation details in Appendix B.1, details of the benchmark settings in Appendix B.2, and details of the hyperparameters in Appendix B.3 to facilitate reproducibility.

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

## A USE OF LLMs DISCLOSURE

We employed GPT-5 (OpenAI, 2025a) from OpenAI to assist with language polishing in order to improve the readability of the paper. We affirm that large language models were not misused intentionally in any part of this work. All intellectual contributions are attributed to the human authors, and the results presented in this paper are entirely the product of human research efforts.

## B SUPPLEMENTARY DETAILS

### B.1 SWIREASONING IMPLEMENTATION

---

**Algorithm 1** SWIREASONING

**Input:** Question $x_{1:n}$, model $\mathcal{M}$, max steps $T_{\max}$, coefficient $\alpha_0$, coefficient $\beta_0$, dwell window $W_{\mathrm{E}\to\mathrm{L}}$, max switches $C_{\max}$, and answer budget $B$
**Output:** Answer $y_{1:m}$
1   **Init:** Mode $m_0 \leftarrow$ Latent, switch counter $C \leftarrow 0$, injection queue $Q \leftarrow \emptyset$, budget flag $b \leftarrow -1$
2   **for** $t = 1$ to $T_{\max}$ **do**
3     $\ell_t \leftarrow \mathcal{M}(x_{1:t-1})$;   $p_t \leftarrow \mathrm{softmax}(\ell_t)$;   $H_t \leftarrow -\sum_v p_t[v] \log p_t[v]$       *// Entropy*
4     **if** $Q \neq \emptyset$ **then**            *// Token injection (convergence/termination prefix)*
5       $x_t \leftarrow Q.\mathrm{pop}()$
6       **if** $b = 0$ **then**
7         **break**
8       **if** $b > 0$ **then**
9         $b \leftarrow b - 1$
10       **continue**
11     **if** $t = 1$ **then**
12       $\bar{H} \leftarrow H_1$;    $\Delta t \leftarrow 0$
13     **if** $m_{t-1} =$ Latent **and** $H_t < \bar{H}$ **then**       *// Mode switching (Sec. 3.3)*
14       $m_t \leftarrow$ Explicit;    $\bar{H} \leftarrow H_t$;    $\Delta t \leftarrow 0$;    $C \leftarrow C + 1$
15     **else if** $m_{t-1} =$ Explicit **and** $H_t > \bar{H}$ **and** $\Delta t \geq W_{\mathrm{E}\to\mathrm{L}}$ **then**
16       $m_t \leftarrow$ Latent;    $\bar{H} \leftarrow H_t$;    $\Delta t \leftarrow 0$
17     **else**
18       $m_t \leftarrow m_{t-1}$;    $\Delta t \leftarrow \Delta t + 1$
19     **if** $m_t =$ Explicit **and** $\frac{1}{2}C_{\max} \leq C \leq C_{\max}$ **then**       *// Switch count control (Sec. 3.4)*
20       $Q \leftarrow [\,\mathrm{ID}[\langle/\mathrm{think}\rangle]\,]$       *// Convergence trigger*
21     **else if** $m_t =$ Explicit **and** $C > C_{\max}$ **then**
22       $Q \leftarrow [\,\mathrm{ID}[\text{``}\langle/\mathrm{think}\rangle\backslash\mathrm{n}\backslash\mathrm{n}\ \text{The final answer is''}]\,]$;    $b \leftarrow B$       *// Termination trigger*
23     **if** $m_t =$ Explicit **and** $\Delta t > 0$ **then**
24       $x_t \leftarrow \arg\max_v p_t[v]$ or Sampling
25     **else**
26       $\tilde{e}_t \leftarrow \sum_v p_t[v]\, E[v]$
27       **if** $m_t =$ Latent **and** $\Delta t = 0$ **then**       *// Thinking-related signal mixing*
28         $\alpha_t = \alpha_0 + (1 - \alpha_0)\frac{t}{T_{\max}}$
29         $\tilde{e}_t \leftarrow \alpha_t\, \tilde{e}_t + (1 - \alpha_t)\, e_{\langle\mathrm{think}\rangle}$
30       **if** $m_t =$ Explicit **and** $\Delta t = 0$ **then**       *// Thinking-related signal mixing*
31         $\beta_t = \beta_0 + (1 - \beta_0)\frac{t}{T_{\max}}$
32         $\tilde{e}_t \leftarrow \beta_t\, \tilde{e}_t + (1 - \beta_t)\, e_{\langle/\mathrm{think}\rangle}$
33       $x_t \leftarrow \tilde{e}_t$       *// Soft embeddings feed as inputs*
34     **if** $x_t = {<}\mathrm{EOS}{>}$ **then**
35       **break**
36   Extract answer $y$ from $x_{n+1:t}$
37   **return** $y$

---

Alg. 1 provides a detailed implementation of SWIREASONING, where the implementation for mode switching is written in black and switch count control for token efficiency is outlined in blue.

## B.2 BENCHMARK DETAILS

We conduct evaluations on 11 reasoning benchmarks, including GSM8K (Cobbe et al., 2021), Math500 (Hendrycks et al., 2021), AIME 2024 (HuggingFaceH4, 2024), AIME 2025 (Yentinglin, 2025) for mathematical reasoning; GPQA Diamond (Rein et al., 2024) for STEM reasoning; HumanEval (Chen et al., 2021), LeetCode-Contest (Guo et al., 2024), MBPP (Austin et al., 2021), LiveCodeBench (Jain et al., 2024) for coding reasoning; and 2WikiMultihopQA (Ho et al., 2020), CommonsenseQA (Talmor et al., 2019) for general reasoning.

- **GSM8K**: We use the test set of 1,319 grade-school math word problems, designed to evaluate multi-step arithmetic reasoning in natural language. 🤗: https://huggingface.co/datasets/openai/gsm8k.

- **MATH500**: A curated set of 500 problems from the MATH dataset, covering diverse high-school competition-level mathematics domains such as algebra, geometry, and number theory. 🤗: https://huggingface.co/datasets/HuggingFaceH4/MATH-500.

- **AIME 2024**: Contains 30 problems from the 2024 American Invitational Mathematics Examination, each requiring a concise numeric answer and reflecting competition-level difficulty. 🤗: https://huggingface.co/datasets/HuggingFaceH4/aime_2024.

- **AIME 2025**: Contains 30 problems from the 2025 American Invitational Mathematics Examination, continuing the focus on competition-style math reasoning with challenging questions that test symbolic and logical skills. 🤗: https://huggingface.co/datasets/yentinglin/aime_2025.

- **GPQA Diamond**: A high-quality subset of about 198 carefully verified questions, focusing on STEM disciplines including mathematics, physics, chemistry, biology, and computer science, designed to evaluate expert-level factual knowledge and reasoning ability. 🤗: https://huggingface.co/datasets/hendrydong/gpqa_diamond_mc.

- **HumanEval**: A set of 164 hand-written Python programming problems. Each problem provides a function signature and a short docstring specification, and models are evaluated by executing generated code against unit tests. 🤗: https://huggingface.co/datasets/openai/openai_humaneval.

- **LeetCode-Contest**: A benchmark for evaluating code LLMs, consisting of 180 algorithm problems of different difficulties from LeetCode contests. The evaluation is based on whether the generated solution passes the provided tests. 🤗: https://huggingface.co/datasets/TechxGenus/LeetCode-Contest.

- **MBPP**: We use the test set of 257 Python programming problems. Each problem includes a natural language prompt, a reference solution, and unit tests. The model is scored by running generated code on the tests. 🤗: https://huggingface.co/datasets/google-research-datasets/mbpp.

- **LiveCodeBench**: The dataset contains 279 problems, and the prompt instructs the model to generate a correct Python program that matches the specification and passes all tests. 🤗: https://huggingface.co/datasets/PrimeIntellect/LiveCodeBench-v5.

- **2WikiMultihopQA**: We use the 2WikiMultihopQA set with 200 problems from Long-Bench (Bai et al., 2024). The problems require combining evidence from multiple Wikipedia documents to reach the final answer. It tests cross-document reasoning rather than single passage lookup. 🤗: https://huggingface.co/datasets/zai-org/LongBench.

- **CommonsenseQA**: We use the test set of 1,221 multiple-choice QA questions that require commonsense knowledge. Questions are constructed from ConceptNet relations and each question comes with one correct answer and several distractor options. 🤗: https://huggingface.co/datasets/tau/commonsense_qa.

To provide LLMs with sufficient thinking space, following the same settings as Qwen3 (Yang et al., 2025), we set the maximum generation length to 32,768 tokens for GSM8K, Math500, GPQA Diamond, HumanEval, LeetCode-Contest, MBPP, LiveCodeBench, 2WikiMultihopQA, and CommonsenseQA benchmarks, and 38,912 tokens for AIME 2024 and AIME 2025 benchmarks.

We repeat the experiments eight times and report the average accuracy for both SWIREASONING and other baselines on the AIME 2024 and AIME 2025 benchmarks.

## B.3 BEST PRACTICE FOR HYPERPARAMETERS

Table 6: Hyperparameters for mode switching across datasets and models. $W$ and $\beta_0$ are fixed across all scenarios, while $\alpha_0$ provides users with flexibility for adjustment depending on the task.

| Hyperparameter | Dataset | Model | | | |
|---|---|---|---|---|---|
| | | *Qwen3-1.7B* | *Qwen3-8B* | *Qwen3-32B* | *DeepSeek-R1-Distill-Llama-8B* |
| $W$ (window size) | GSM8K
MATH500
AIME2024
AIME2025
GPQA Diamond
HumanEval
LeetCode-Contest
MBPP
LiveCodeBench
2WikiMultihopQA
CommonsenseQA | | | 512 (fixed for all) | |
| $\alpha_0$ (user-exposed) | GSM8K
MATH500
AIME2024
AIME2025
GPQA Diamond
HumanEval
LeetCode-Contest
MBPP
LiveCodeBench
2WikiMultihopQA
CommonsenseQA | 0.6
0.5
0.5
0.3
1.0


/


 | 0.5
1.0
0.9
0.9
1.0
0.5
1.0
0.9
0.9
0.7
0.9 | 0.5
0.5
1.0
0.9
0.6


/


 | 0.1
0.5
0.65
0.7
0.7


/


 |
| $\beta_0$ | GSM8K
MATH500
AIME2024
AIME2025
GPQA Diamond
HumanEval
LeetCode-Contest
MBPP
LiveCodeBench
2WikiMultihopQA
CommonsenseQA | | | 0.7 (fixed for all) | |

In addition to Tab. 6, SWIREASONING operates as a straightforward and instant substitution for the `model.generate()` interface of Huggingface's *transformers* (Wolf et al., 2020) package. There are no model parameters or architecture changes, and no training was used in the experiments. For sampling-related hyperparameters and prompt templates, we use the ones recommended by Qwen3 and DeepSeek-R1's technical report (Yang et al., 2025; Guo et al., 2025) without modification.

## B.4 BROADER RELATED WORK

**Efficient LLM Reasoning.** In terms of improving reasoning efficiency, there are broader techniques including but not limited to KV cache compression (Han et al., 2023; Xiao et al., 2023; Cai et al., 2024; Shi et al., 2025), prompt token compression (Wingate et al., 2022; Jiang et al., 2023; Pan et al., 2024; Shi et al., 2024), speculative decoding (Leviathan et al., 2023; Kim et al., 2023; Liu et al., 2023; Sun et al., 2023; Chen et al., 2024), traditional methods such as quantization, pruning, distillation (Shi et al., 2023; Lin et al., 2024; Fu et al., 2024; Yuan et al., 2025b;a), and system-level optimizations such as FlashAttention (Dao et al., 2022; Dao, 2023; Shah et al., 2024). Learning-

based generation methods, such as COCONUT (Hao et al., 2024) based on supervised fine-tuning, and Long⊗Short (Ning et al., 2025) based on reinforcement learning, shorten the reasoning process by involving additional training.

SwiReasoning, however, targets a different axis of efficiency and is not aiming to surpass them. Instead, it saves tokens by dynamically alternating between latent steps and explicit steps and limiting the number of block switches. As such, it is plug-and-play during inference and can be layered on top of the aforementioned techniques for multiplicative gains.

## C SUPPLEMENTARY EXPERIMENTS

### C.1 QUANTITATIVE ANALYSES OF EXPLORATION IN LATENT AND EXPLICIT MODES

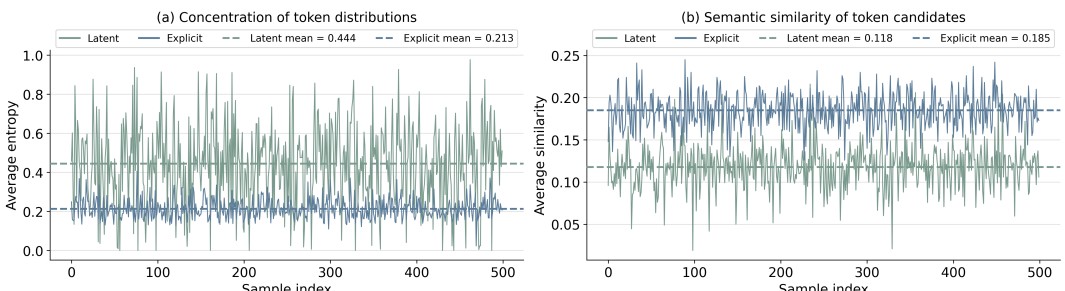

Figure 6: Latent modes enable exploration with less concentrated token distributions and more semantically diverse candidates. In latent modes: (a) the average entropy of the next-token distributions indicates less concentration, and (b) the average pairwise similarity among token candidates indicates more semantic diversity.

To quantify exploration in the latent and explicit modes, we analyze two complementary statistics with the Qwen3-8B model on Math500 as visualized in Fig. 6. First, to measure the differences in the concentration level of the next-token distributions between latent and explicit modes, we calculate the average Shannon entropy across steps for each sample. Fig. (a) shows that the entropy in latent mode is 2.08x higher than in explicit mode on average (0.444 vs. 0.213), indicating that probability mass is spread across more candidate tokens instead of collapsing in latent mode.

Entropy alone does not guarantee semantic diversity, as the semantic distances of different token candidates can be close, even when the probability mass is spread across them. To measure the differences in the semantic similarity among token candidates between latent and explicit modes, we calculate the average pairwise cosine similarity of top-5 token candidates over steps by using their embeddings for each sample. Fig. (b) shows that the semantic similarity in latent mode is 63.8% of that in explicit mode on average (0.118 vs. 0.185), showing that token candidates considered under latent modes are more semantically diverse. Taken together, these results quantitatively substantiate that latent modes enable exploration with less concentrated token distributions and more semantically diverse candidates.

### C.2 EFFICIENCY MEASURED IN TERMS OF LATENCY AND TFLOPS

We evaluate efficiency in terms of average wall-clock latency and TFLOPs over samples on MATH500 with Qwen3-8B, comparing CoT, CoT (Greedy), Soft Thinking, and SwiReasoning. Fig. 7 demonstrates that SwiReasoning consistently reaches a given accuracy with lower latency and fewer TFLOPs, and the advantage grows as the budget becomes tighter. Specifically, at 90% pass@1 accuracy, SwiReasoning uses 1.36× fewer TFLOPs and is 1.36× faster in wall-clock time than CoT. At 80% accuracy, it uses 2.18× fewer TFLOPs and is 2.17× faster. These results confirm that SwiReasoning improves efficiency at fixed accuracies, especially under small budgets.

### C.3 STATISTICS OF SWITCH COUNTS ACROSS PROBLEM DIFFICULTIES

To investigate the switching dynamics, we analyzed the distribution of switch counts across five benchmarks of varying difficulty. Fig. 8 presents the box plots of switch counts for Qwen3-1.7B,

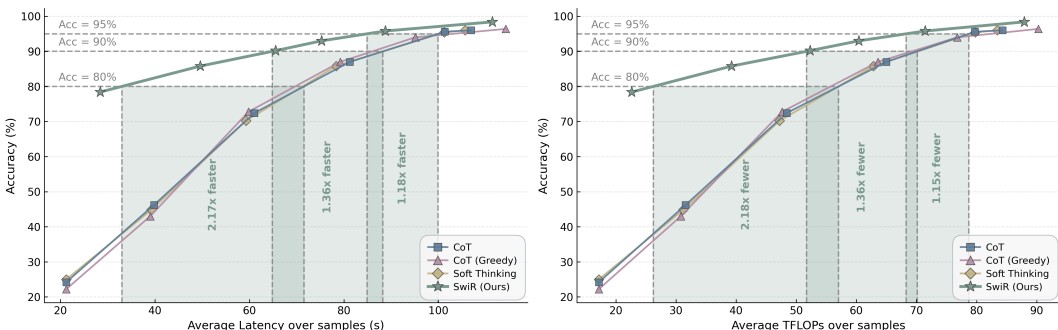

Figure 7: Left: Accuracy vs. wall-clock latency. Right: Accuracy vs. TFLOPs. Dashed horizontal lines mark equal-accuracy targets (80%, 90%, 95%). Lower values on the x-axes are better.

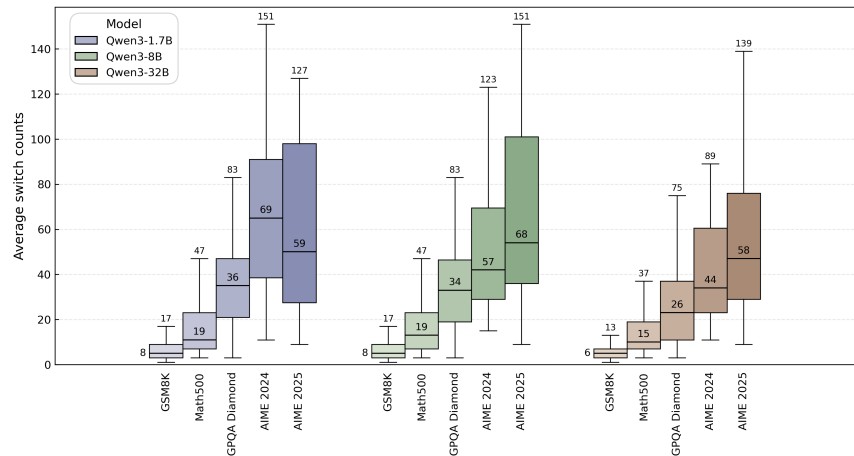

Figure 8: Statistics of switch counts across varying problem difficulties and model scales. The box plots show the distribution of the number of switches for Qwen3-1.7B, Qwen3-8B, and Qwen3-32B on five benchmarks.

Qwen3-8B, and Qwen3-32B. The switching frequency demonstrates adaptability to problem difficulty. On benchmarks requiring complex reasoning such as AIME 2024 and AIME 2025, the switch counts increase substantially, with medians ranging from 44 to 69, and maximum counts reach 151. The results confirm that SwiReasoning functions as a difficulty-aware reasoning approach.

## C.4 ANALYSIS OF CHALLENGES IN SPECIFIC SCENARIOS

### 3D Surface Shortest Path

▶**Question:** A gecko is in a room that is 12 feet long, 10 feet wide and 8 feet tall. The gecko is currently on a side wall ($10'$ by $8'$), one foot from the ceiling and one foot from the back wall ($12'$ by $8'$). The gecko spots a fly on the opposite side wall, one foot from the floor and one foot from the front wall. What is the length of the shortest path the gecko can take to reach the fly assuming that it does not jump and can only walk across the ceiling and the walls? Express your answer in simplest radical form.

▶**Output:**
To determine the shortest path the gecko can take to reach the fly, we consider the constraints:

- The **gecko** is on a **side wall** (10' by 8') of a **12' × 10' × 8'** room.
- The **fly** is on the **opposite side wall**.
- The gecko is **one foot from the ceiling** and **one foot from the back wall**.
- The fly is **one foot from the floor** and **one foot from the front wall**.

- The gecko **can only walk across the ceiling and walls**, not the floor.

---

STEP 1: ASSIGN COORDINATES

Let's define the room in 3D coordinates:

- The **length** (along the x-axis) is 12 feet.
- The **width** (along the y-axis) is 10 feet.
- The **height** (along the z-axis) is 8 feet.

Assume:

- The **gecko** is on the **left wall** (x = 0), one foot from the **back wall** (y = 1), and one foot from the **ceiling** (z = 7). So its position is:
  **(x = 0, y = 1, z = 7)**
- The **fly** is on the **right wall** (x = 12), one foot from the **front wall** (y = 9), and one foot from the **floor** (z = 1). So its position is:
  **(x = 12, y = 9, z = 1)**

---

STEP 2: CONSIDER THE SHORTEST PATH

The gecko can only walk **across the ceiling and walls**. The shortest path will involve **unfolding** the room into a flat 2D plane such that the gecko's path becomes a straight line.
A common and effective approach is to **unfold** the **left wall**, **ceiling**, and **right wall** into a single plane. This allows the gecko to move from its position on the left wall, across the ceiling, to the fly on the right wall.

---

STEP 3: TRANSFORM TO A 2D COORDINATE SYSTEM

In this unfolded net:

- The **left wall** is 10 feet wide (y-axis) and 8 feet tall (z-axis).
- The **ceiling** is 12 feet long (x-axis) and 10 feet wide (y-axis).
- The **right wall** is 10 feet wide (y-axis) and 8 feet tall (z-axis).

When we unfold the left wall, ceiling, and right wall into one plane, the gecko's position and the fly's position can be mapped in this 2D plane.
In this 2D representation:

- The **gecko's position** is at (y = 1, z = 7) on the left wall.
- The **fly's position** is at (y = 9, z = 1) on the right wall.

However, since the gecko is on the **left wall**, and the fly is on the **right wall**, and the path goes **across the ceiling**, we can consider the **horizontal distance** as the **length of the room (12 feet)**, and the **vertical distance** as the **difference in heights from the gecko to the ceiling (1 foot)** and from the ceiling to the fly (7 feet), totaling **8 feet**.
So, the total path forms a **right triangle** with: [Incorrect reasoning trace here that does not adhere to a "only walk across the ceiling and the walls" constraint]

- One leg: **12 feet** (length of the room)
- The other leg: **8 feet** (vertical distance from gecko to ceiling + ceiling to fly)

---

STEP 4: COMPUTE THE SHORTEST DISTANCE

Using the Pythagorean theorem:

$$\text{Distance} = \sqrt{12^2 + 8^2} = \sqrt{144 + 64} = \sqrt{208} = \sqrt{16 \cdot 13} = 4\sqrt{13}$$

---

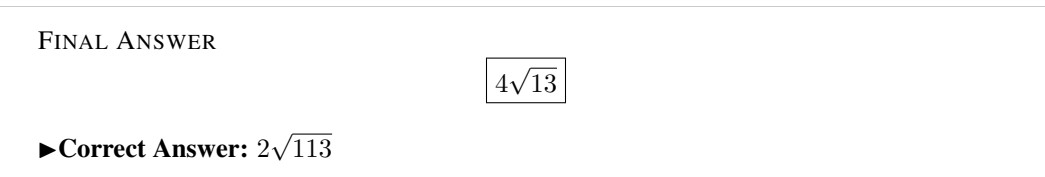

FINAL ANSWER

$$\boxed{4\sqrt{13}}$$

▶**Correct Answer:** $2\sqrt{113}$

While SWIREASONING demonstrates robust generalization across diverse domains, we identify a specific category of tasks, spatial reasoning with rigid topological constraints, which may be challenging for latent-based methods. A representative case is the "3D Surface Shortest Path" problem from the Math500 benchmark shown above. In this problem, a gecko must find the shortest path on a 3D room's surface. The reasoning trace reveals that SWIREASONING correctly identifies the high-level strategy: unfolding the 3D room into a 2D plane. However, it struggles to maintain the specific topological "only walk across the ceiling and the walls" constraint during latent transitions.

We attribute this behavior to a trade-off inherent in latent reasoning:

- Smoothing vs. Precision: Latent reasoning operates by smoothing probability distributions to enable diverse semantic exploration. While this is beneficial for logical reasoning, it may be detrimental for tasks with precise constraints, such as rigid geometric topology (*e.g.*, defining exactly which wall edge connects to which).
- Blurring Constraints: During the latent mode, the strict "only walk across the ceiling and the walls" constraint is blurred due to the probability mass spreading. As a result, when the model switches back to explicit mode, it may reason with an invalid path (*e.g.*, calculating a direct Euclidean distance through prohibited space), leading to an incorrect answer ($4\sqrt{13}$ vs. true $2\sqrt{113}$).

Overall, this suggests that while SWIREASONING excels at semantic and logical exploration, a few tasks requiring rigid geometric constraint satisfaction, where fuzzy latent exploration may interfere with precise geometric execution, could still be challenging.

## C.5 ABLATION STUDY ON THE CONVERGENCE TRIGGERS W.R.T. TERMINATION ONES

Table 7: Ablation study on the relative position of convergence and termination triggers. The convergence trigger is set at different fractions of the termination count $T$ (where $T = C_{\max}$). The results demonstrate that the default $\frac{1}{2}T$ offers a balanced choice, avoiding the significant accuracy loss of earlier triggers while being more efficient than later ones.

| **Convergence w.r.t. Termination** | $C = \frac{1}{4}T$ | $C = \frac{1}{3}T$ | $C = \frac{1}{2}T$ | $C = \frac{2}{3}T$ | $C = \frac{3}{4}T$ |
|---|---|---|---|---|---|
| $C_{\max} = 8$ | | | | | |
| Accuracy (%) | 76.60 | 79.20 | 81.00 | 84.20 | 84.20 |
| Generation Length | 1428 | 1817 | 2222 | 2743 | 2743 |
| $C_{\max} = 20$ | | | | | |
| Accuracy (%) | 84.00 | 85.80 | 87.80 | 88.20 | 88.80 |
| Generation Length | 2788 | 3278 | 3597 | 4002 | 3943 |

To analyze the positions of convergence triggers w.r.t. termination ones, we conducted an ablation study using Qwen3-1.7B on the Math500 benchmark. We fixed the termination trigger at two representative budgets ($C_{\max} = 8$ and $C_{\max} = 20$) and varied the position of the convergence trigger relative to the termination step. Specifically, if $T$ denotes the switch count at termination (*i.e.*, $T = C_{\max}$), we tested setting the convergence trigger at $C \in \{\frac{1}{4}T, \frac{1}{3}T, \frac{1}{2}T, \frac{2}{3}T, \frac{3}{4}T\}$. Tab. 7 presents a clear trade-off between accuracy and token efficiency:

- Early convergence ($\frac{1}{4}T, \frac{1}{3}T$): triggering convergence too early significantly reduces generation length but leads to a notable drop in accuracy (*e.g.*, at $C_{\max} = 20$, accuracy drops from $87.80\%$ to $84.00\%$ when moving from $\frac{1}{2}T$ to $\frac{1}{4}T$). This suggests that forcing the model to converge before sufficient exploration prevents it from finding correct solutions for harder problems.
- Late convergence ($\frac{2}{3}T, \frac{3}{4}T$): delaying the convergence trigger yields marginal accuracy gains (*e.g.*, +1.0% from $\frac{1}{2}T$ to $\frac{3}{4}T$ at $C_{\max} = 20$) but comes at the cost of increased token consumption.

- Balanced default ($\frac{1}{2}T$): the default setting of $\frac{1}{2}C_{\max}$ effectively strikes a balance. It captures the majority of the accuracy while maintaining reasonable token efficiency. Overall, $C = \frac{1}{2}T$ serves as a sweet spot for general-purpose reasoning under budget constraints.

## C.6 ABLATION STUDY ON ISOLATING THE EFFECT OF LATENT UNDER SMALL BUDGETS

Table 8: Ablation study on isolating the effect of latent under small token budgets. "SwiR w/o latent" retains the switch count controller but disables latent mode.

| Method | GSM8K | MATH 500 | GPQA Diamond | AIME 2024 | AIME 2025 | Average | |
|---|---|---|---|---|---|---|---|
| | | | *Accuracy (%)* | | | | |
| SwiR w/o latent | 85.44 | 68.60 | 26.26 | 2.50 | 0.42 | 36.64 | +0.00 |
| *SwiR* | **86.80** +1.36 | **72.40** +3.80 | **29.80** +3.54 | **5.83** +3.33 | **4.58** +4.16 | **39.88** +3.24 |
| | | | *Generation Length* | | | | |
| SwiR w/o latent | **816** | 1207 | 1032 | 835 | 791 | 936 | +0.00 |
| *SwiR* | **816** | **1189** | **933** | **812** | **787** | **907** | −3.10% |

To further evaluate the effect of latent modes at small budgets, we used Qwen3-1.7B and compared the full SWIREASONING against a variant where latent reasoning is disabled (SwiR w/o latent). In the "w/o latent" setting, the model is forced to remain in explicit mode, but retains the same switch count controller (functioning as a step-limit controller for inserting the proposed convergence and termination triggers). We evaluated both methods under small token budgets.

Tab. 8 shows that disabling latent leads to a consistent degradation in performance across all benchmarks. Under lower or the same token usage, the full SwiReasoning achieves an average accuracy improvement of +3.24% over the ablation. On challenging benchmarks like AIME2025, the contribution of latent is particularly pronounced (+4.16%). These results show that the latent itself provides gains that cannot be attributed to token controller mechanisms alone.

## C.7 SUPPLEMENTARY COMPONENT-WISE ABLATION STUDY

Table 9: Supplementary ablation studies on switch window mechanisms (including window size, symmetry vs. asymmetry) and signal mixing.

| Method | GSM8K | Math500 | Average |
|---|---|---|---|
| *SwiR* w/ window size 0 | 89.00% | 89.20% | 89.10% |
| *SwiR* w/ symmetry window size 2 | 89.31% | 90.00% | 89.66% |
| *SwiR* w/ symmetry window size 4 | **90.83%** | 89.20% | 90.02% |
| *SwiR* w/ symmetry window size 8 | 90.60% | 91.20% | 90.90% |
| *SwiR* w/ symmetry window size 16 | 88.93% | 92.00% | 90.47% |
| *SwiR* w/ symmetry window size 32 | 90.14% | 91.00% | 90.57% |
| *SwiR* w/ asymmetry window size 64 | 89.69% | 92.60% | 91.15% |
| *SwiR* w/ asymmetry window size 128 | 90.45% | 91.00% | 90.72% |
| *SwiR* w/ asymmetry window size 256 | 89.76% | 90.80% | 90.28% |
| *SwiR* w/ asymmetry window size = 512 ✓ | **90.83%** | **93.00%** | **91.92%** |
| *SwiR* w/ asymmetry window size = 1024 | **90.83%** | 91.20% | 91.01% |
| *SwiR* w/o mixing | 90.44% | 92.20% | 91.32% |

To quantify the marginal contribution of each proposed mechanism, we conducted a component-wise ablation study using Qwen3-1.7B on GSM8K and MATH500 benchmarks. We compared the default asymmetric window ($W_{E \to L} = 512, W_{L \to E} = 0$) against variations with no window ($W = 0$) and symmetric windows of varying sizes. Results in Tab. 9 reveal that

- Necessity of a window: Removing the window constraint completely leads to a significant performance drop (Avg: $89.10\%$ vs. $91.92\%$), due to rapid oscillations between modes preventing coherent reasoning chains from forming.

- Benefit of asymmetry: The default asymmetric setting outperforms all symmetric configurations. While symmetric windows (*e.g.*, size 8) improve over no window, they fail to match the peak performance of the asymmetric design. This validates our hypothesis that explicit reasoning requires a consolidation period ($W_{E \to L}$), while latent reasoning benefits from immediate exit upon confidence recovery ($W_{L \to E} = 0$) to reduce the risks of introducing spurious signals that may mislead the model.

Tab. 9 also indicates that disabling the signal mixing results in a consistent accuracy decrease (Avg: $91.32\%$ vs. $91.92\%$). This suggests that the signal mixing helps the model better orient its internal state during mode transitions and contributes positively to the final accuracy.

Table 10: Supplementary ablation study on the maximum switch count $C_{\max}$. We evaluate the trade-off between accuracy and generation length by varying $C_{max}$ from 4 to unlimited.

| $\mathbf{C}_{\max}$ | 4 | 8 | 12 | 16 | 20 | 24 |
|---|---|---|---|---|---|---|
| Accuracy | 72.40% | 81.00% | 85.00% | 86.80% | 87.80% | 87.80% |
| Generation Length | 1189 | 2222 | 2914 | 3280 | 3597 | 3862 |
| $\mathbf{C}_{\max}$ | 28 | 32 | 36 | 40 | 44 | $\infty$ |
| Accuracy | 89.00% | 89.20% | 89.80% | 90.60% | 91.80% | **93.00%** |
| Generation Length | 4068 | 4186 | 4274 | 4318 | 4602 | 4924 |

We analyzed the effect of the switch count cap ($C_{max}$) on accuracy and generation length using Qwen3-1.7B on MATH500 in Tab. 10. The results indicate that

- Efficiency control: As $C_{max}$ decreases (from $\infty$ down to 4), the generation length drops significantly (from 4924 to 1189), demonstrating the controller's effectiveness in curbing overthinking. This provides users with sufficient capabilities to control the intensity of thinking flexibly.

- Accuracy trade-off: While extremely tight caps (*e.g.*, 4 or 8) limit accuracy, moderate caps (*e.g.*, 20-32) achieve performance competitive with or close to the unlimited setting ($C_{max} = \infty$) but with much lower token consumption. This confirms that the controller effectively makes use of partial reasoning trajectories to deliver reasonable answers.

## C.8 DETAILED EVALUATION RESULTS UNDER VARYING TOKEN BUDGETS

We provide detailed evaluation results of Qwen3-8B in Tab. 11-15, Qwen3-1.7B in Tab. 16-20, and DeepSeek-R1-Distill-Llama-8B in Tab. 21-25.

Table 11: Evaluation results of Qwen3-8B on the GSM8K benchmark under varying token budgets. Rows are *sorted by generation length in descending order*.

| Method | Accuracy (%) | Generation Length |
|---|---|---|
| CoT (Greedy) | 95.68% | 2240 |
| SwiR (Ours) | 96.06% | 2218 |
| CoT (Greedy) | 95.75% | 2199 |
| CoT | 95.60% | 2138 |
| CoT | 95.60% | 2136 |
| CoT | 94.77% | 2123 |
| CoT (Greedy) | 95.15% | 2115 |
| Soft Thinking | 95.38% | 2073 |
| Soft Thinking | 95.07% | 2033 |
| CoT (Greedy) | 92.65% | 1934 |
| CoT | 91.81% | 1926 |
| SwiR (Ours) | 94.84% | 1879 |
| Soft Thinking | 92.12% | 1865 |
| SwiR (Ours) | 95.14% | 1761 |
| SwiR (Ours) | 94.39% | 1585 |
| CoT | 79.90% | 1553 |
| CoT (Greedy) | 79.68% | 1540 |
| Soft Thinking | 80.14% | 1526 |
| SwiR (Ours) | 94.47% | 1297 |
| CoT | 44.50% | 990 |
| CoT (Greedy) | 45.79% | 988 |
| Soft Thinking | 47.08% | 988 |
| SwiR (Ours) | 93.70% | 844 |
| CoT | 25.47% | 512 |
| CoT (Greedy) | 25.93% | 512 |
| Soft Thinking | 24.87% | 512 |
| SwiR (Ours) | 92.19% | 301 |
| CoT | 6.36% | 256 |
| CoT (Greedy) | 6.07% | 256 |
| Soft Thinking | 6.22% | 256 |

Table 12: Evaluation results of Qwen3-8B on the MATH500 benchmark under varying token budgets. Rows are *sorted by generation length in descending order*.

| Method | Accuracy (%) | Generation Length |
|---|---|---|
| CoT (Greedy) | 96.40% | 5311 |
| SwiR (Ours) | 98.40% | 5183 |
| CoT | 96.00% | 4985 |
| Soft Thinking | 96.00% | 4934 |
| Soft Thinking | 95.40% | 4733 |
| CoT | 95.60% | 4729 |
| CoT (Greedy) | 94.00% | 4565 |
| SwiR (Ours) | 95.80% | 4266 |
| SwiR (Ours) | 93.80% | 4057 |
| CoT | 87.00% | 3899 |
| CoT (Greedy) | 87.00% | 3819 |
| Soft Thinking | 85.80% | 3774 |
| SwiR (Ours) | 93.00% | 3635 |
| SwiR (Ours) | 90.20% | 3164 |
| CoT | 72.40% | 2940 |
| CoT (Greedy) | 72.80% | 2890 |
| Soft Thinking | 70.20% | 2865 |
| SwiR (Ours) | 85.80% | 2387 |
| CoT | 46.20% | 1922 |
| Soft Thinking | 44.80% | 1898 |
| CoT (Greedy) | 43.00% | 1873 |
| SwiR (Ours) | 78.40% | 1368 |
| CoT | 24.20% | 1024 |
| Soft Thinking | 25.00% | 1024 |
| CoT (Greedy) | 22.20% | 1023 |

Table 13: Evaluation results of Qwen3-8B on the GPQA Diamond benchmark under varying token budgets. Rows are *sorted by generation length in descending order*.

| Method | Accuracy (%) | Generation Length |
|---|---|---|
| **SwiR (Ours)** | 61.11% | 8359 |
| Soft Thinking | 59.59% | 8153 |
| CoT | 59.60% | 8123 |
| CoT (Greedy) | 56.57% | 7909 |
| CoT | 55.56% | 7570 |
| CoT (Greedy) | 55.05% | 7546 |
| Soft Thinking | 55.05% | 7433 |
| **SwiR (Ours)** | 58.08% | 7100 |
| **SwiR (Ours)** | 57.07% | 6338 |
| **SwiR (Ours)** | 58.08% | 5710 |
| CoT (Greedy) | 33.84% | 5086 |
| CoT | 33.33% | 4972 |
| Soft Thinking | 34.85% | 4961 |
| **SwiR (Ours)** | 55.05% | 4766 |
| **SwiR (Ours)** | 53.54% | 3603 |
| CoT (Greedy) | 12.12% | 3078 |
| Soft Thinking | 12.63% | 2959 |
| CoT | 10.61% | 2861 |
| **SwiR (Ours)** | 46.96% | 2117 |
| Soft Thinking | 2.53% | 1753 |
| CoT | 2.52% | 1743 |
| CoT (Greedy) | 2.02% | 1723 |
| **SwiR (Ours)** | 47.47% | 1527 |
| CoT | 0.00% | 1024 |
| CoT (Greedy) | 0.00% | 1024 |
| Soft Thinking | 0.00% | 1024 |
| **SwiR (Ours)** | 39.39% | 867 |
| CoT | 0.00% | 512 |
| CoT (Greedy) | 0.00% | 512 |
| Soft Thinking | 0.00% | 512 |

Table 14: Evaluation results of Qwen3-8B on the AIME2024 benchmark under varying token budgets. Rows are *sorted by generation length in descending order*.

| Method | Accuracy (%) | Generation Length |
|---|---|---|
| **SwiR (Ours)** | 79.17% | 12491 |
| Soft Thinking | 67.92% | 12271 |
| CoT | 75.83% | 12077 |
| CoT (Greedy) | 70.00% | 11680 |
| **SwiR (Ours)** | 69.58% | 10815 |
| **SwiR (Ours)** | 66.25% | 10349 |
| CoT (Greedy) | 66.67% | 10328 |
| Soft Thinking | 62.92% | 9846 |
| CoT | 63.75% | 9818 |
| **SwiR (Ours)** | 61.25% | 9275 |
| **SwiR (Ours)** | 57.08% | 8115 |
| Soft Thinking | 36.67% | 7343 |
| CoT | 38.75% | 7109 |
| CoT (Greedy) | 36.67% | 7033 |
| **SwiR (Ours)** | 45.42% | 6093 |
| CoT | 20.83% | 4096 |
| Soft Thinking | 23.33% | 4096 |
| CoT (Greedy) | 26.67% | 4056 |
| **SwiR (Ours)** | 25.42% | 3589 |
| CoT | 5.83% | 2048 |
| CoT (Greedy) | 10.00% | 2048 |
| Soft Thinking | 3.75% | 2048 |
| **SwiR (Ours)** | 12.08% | 1809 |
| CoT | 1.67% | 1024 |
| CoT (Greedy) | 3.33% | 1024 |
| Soft Thinking | 3.33% | 1024 |
| **SwiR (Ours)** | 6.67% | 818 |
| CoT | 0.83% | 512 |
| CoT (Greedy) | 0.00% | 512 |
| Soft Thinking | 3.33% | 512 |

Table 15: Evaluation results of Qwen3-8B on the AIME2025 benchmark under varying token budgets. Rows are *sorted by generation length in descending order*.

| Method | Accuracy (%) | Generation Length |
|---|---|---|
| **SwiR (Ours)** | 70.00% | 13911 |
| Soft Thinking | 68.33% | 13665 |
| CoT (Greedy) | 60.00% | 13292 |
| CoT | 67.50% | 12924 |
| **SwiR (Ours)** | 62.92% | 11482 |
| **SwiR (Ours)** | 58.33% | 10596 |
| CoT | 54.17% | 10215 |
| Soft Thinking | 51.25% | 9952 |
| **SwiR (Ours)** | 56.25% | 9791 |
| CoT (Greedy) | 43.33% | 9143 |
| **SwiR (Ours)** | 46.25% | 8220 |
| CoT | 34.58% | 6887 |
| Soft Thinking | 36.25% | 6772 |
| CoT (Greedy) | 33.33% | 6768 |
| **SwiR (Ours)** | 34.58% | 6243 |
| CoT (Greedy) | 13.33% | 4096 |
| CoT | 13.33% | 4091 |
| Soft Thinking | 14.17% | 4060 |
| **SwiR (Ours)** | 21.67% | 3608 |
| CoT | 7.50% | 2048 |
| CoT (Greedy) | 6.67% | 2048 |
| Soft Thinking | 6.25% | 2048 |
| **SwiR (Ours)** | 11.25% | 1999 |
| CoT | 1.67% | 1024 |
| CoT (Greedy) | 0.00% | 1024 |
| Soft Thinking | 3.33% | 1024 |
| **SwiR (Ours)** | 6.67% | 722 |
| CoT | 2.50% | 512 |
| CoT (Greedy) | 0.00% | 512 |
| Soft Thinking | 3.33% | 512 |

Table 16: Evaluation results of Qwen3-1.7B on the GSM8K benchmark under varying token budgets. Rows are *sorted by generation length in descending order*.

| Method | Accuracy (%) | Generation Length |
|---|---|---|
| CoT (Greedy) | 89.61% | 2038 |
| SwiR (Ours) | 90.83% | 2010 |
| CoT | 90.45% | 1981 |
| CoT (Greedy) | 89.61% | 1968 |
| Soft Thinking | 90.30% | 1959 |
| Soft Thinking | 90.22% | 1946 |
| CoT | 89.23% | 1928 |
| Soft Thinking | 89.84% | 1896 |
| CoT (Greedy) | 89.31% | 1895 |
| CoT | 86.35% | 1753 |
| CoT (Greedy) | 87.64% | 1744 |
| Soft Thinking | 87.49% | 1736 |
| SwiR (Ours) | 89.23% | 1695 |
| SwiR (Ours) | 89.46% | 1621 |
| SwiR (Ours) | 87.95% | 1462 |
| CoT | 76.65% | 1420 |
| Soft Thinking | 78.92% | 1418 |
| CoT (Greedy) | 78.17% | 1407 |
| SwiR (Ours) | 88.32% | 1229 |
| CoT | 50.57% | 967 |
| CoT (Greedy) | 48.52% | 959 |
| Soft Thinking | 50.95% | 958 |
| SwiR (Ours) | 86.80% | 816 |
| CoT | 29.95% | 512 |
| CoT (Greedy) | 30.02% | 512 |
| Soft Thinking | 31.54% | 512 |
| SwiR (Ours) | 82.26% | 296 |
| CoT | 7.96% | 256 |
| CoT (Greedy) | 8.34% | 256 |
| Soft Thinking | 9.25% | 256 |

Table 17: Evaluation results of Qwen3-1.7B on the MATH500 benchmark under varying token budgets. Rows are *sorted by generation length in descending order*.

| Method | Accuracy (%) | Generation Length |
|---|---|---|
| SwiR (Ours) | 93.00% | 4924 |
| CoT (Greedy) | 91.00% | 4799 |
| CoT | 92.00% | 4780 |
| Soft Thinking | 90.60% | 4721 |
| CoT | 90.80% | 4435 |
| CoT (Greedy) | 89.20% | 4342 |
| Soft Thinking | 89.00% | 4288 |
| SwiR (Ours) | 87.80% | 3862 |
| CoT (Greedy) | 83.60% | 3681 |
| CoT | 83.60% | 3655 |
| Soft Thinking | 83.60% | 3605 |
| SwiR (Ours) | 87.80% | 3597 |
| SwiR (Ours) | 86.80% | 3280 |
| SwiR (Ours) | 85.00% | 2914 |
| CoT | 68.40% | 2761 |
| Soft Thinking | 69.40% | 2744 |
| CoT (Greedy) | 69.20% | 2738 |
| SwiR (Ours) | 81.00% | 2222 |
| CoT | 46.40% | 1857 |
| CoT (Greedy) | 47.80% | 1850 |
| Soft Thinking | 46.20% | 1830 |
| SwiR (Ours) | 72.40% | 1189 |
| CoT | 25.60% | 1022 |
| CoT (Greedy) | 29.80% | 1020 |
| Soft Thinking | 27.20% | 1020 |

Table 18: Evaluation results of Qwen3-1.7B on the GPQA Diamond benchmark under varying token budgets. Rows are *sorted by generation length in descending order*.

| Method | Accuracy (%) | Generation Length |
|---|---|---|
| **SwiR (Ours)** | 41.41% | 9517 |
| CoT (Greedy) | 31.82% | 9190 |
| CoT (Greedy) | 30.30% | 8751 |
| Soft Thinking | 34.34% | 8731 |
| CoT | 39.39% | 8625 |
| CoT | 37.37% | 8040 |
| **SwiR (Ours)** | 36.87% | 7773 |
| Soft Thinking | 32.32% | 7447 |
| **SwiR (Ours)** | 35.35% | 6792 |
| **SwiR (Ours)** | 37.88% | 5856 |
| **SwiR (Ours)** | 36.87% | 4766 |
| CoT | 17.17% | 4758 |
| Soft Thinking | 16.67% | 4463 |
| CoT (Greedy) | 12.12% | 3770 |
| **SwiR (Ours)** | 37.37% | 3497 |
| Soft Thinking | 8.08% | 2915 |
| CoT | 8.59% | 2843 |
| CoT (Greedy) | 7.58% | 2344 |
| **SwiR (Ours)** | 31.31% | 2112 |
| CoT | 1.01% | 1661 |
| **SwiR (Ours)** | 27.27% | 1573 |
| CoT (Greedy) | 2.53% | 1539 |
| Soft Thinking | 0.51% | 1378 |
| CoT | 0.00% | 1024 |
| CoT (Greedy) | 0.00% | 1024 |
| Soft Thinking | 0.00% | 1024 |
| **SwiR (Ours)** | 29.80% | 933 |

Table 19: Evaluation results of Qwen3-1.7B on the AIME2024 benchmark under varying token budgets. Rows are *sorted by generation length in descending order*.

| Method | Accuracy (%) | Generation Length |
|---|---|---|
| CoT (Greedy) | 40.00% | 12825 |
| **SwiR (Ours)** | 50.83% | 12702 |
| CoT | 45.83% | 11896 |
| Soft Thinking | 38.75% | 10788 |
| **SwiR (Ours)** | 42.08% | 10243 |
| Soft Thinking | 36.67% | 9841 |
| CoT | 38.33% | 9510 |
| CoT (Greedy) | 33.33% | 9377 |
| **SwiR (Ours)** | 38.75% | 9350 |
| **SwiR (Ours)** | 36.25% | 8654 |
| Soft Thinking | 30.00% | 7498 |
| CoT (Greedy) | 23.33% | 7302 |
| **SwiR (Ours)** | 29.17% | 7084 |
| CoT | 27.50% | 6978 |
| **SwiR (Ours)** | 25.42% | 5926 |
| CoT (Greedy) | 10.00% | 4096 |
| Soft Thinking | 16.67% | 4096 |
| CoT | 13.75% | 4063 |
| **SwiR (Ours)** | 14.17% | 3411 |
| CoT | 2.92% | 2048 |
| CoT (Greedy) | 0.00% | 2048 |
| Soft Thinking | 3.33% | 2048 |
| **SwiR (Ours)** | 7.50% | 1887 |
| CoT | 1.25% | 1024 |
| CoT (Greedy) | 0.00% | 1024 |
| Soft Thinking | 0.00% | 1024 |
| **SwiR (Ours)** | 5.83% | 812 |

Table 20: Evaluation results of Qwen3-1.7B on the AIME2025 benchmark under varying token budgets. Rows are *sorted by generation length in descending order*.

| Method | Accuracy (%) | Generation Length |
|---|---|---|
| CoT (Greedy) | 33.33% | 11408 |
| SwiR (Ours) | 38.33% | 9944 |
| CoT | 33.33% | 9733 |
| Soft Thinking | 36.67% | 8904 |
| CoT (Greedy) | 26.67% | 8890 |
| CoT | 31.25% | 8618 |
| SwiR (Ours) | 34.58% | 8543 |
| SwiR (Ours) | 32.92% | 8129 |
| Soft Thinking | 33.33% | 7630 |
| SwiR (Ours) | 32.08% | 7563 |
| SwiR (Ours) | 30.83% | 6761 |
| CoT | 20.83% | 6071 |
| CoT (Greedy) | 20.00% | 6008 |
| Soft Thinking | 23.33% | 5738 |
| SwiR (Ours) | 25.42% | 5145 |
| CoT | 14.17% | 4096 |
| CoT (Greedy) | 13.33% | 3927 |
| Soft Thinking | 20.00% | 3737 |
| SwiR (Ours) | 17.50% | 3311 |
| CoT | 7.92% | 2048 |
| CoT (Greedy) | 10.00% | 2048 |
| Soft Thinking | 10.00% | 2048 |
| SwiR (Ours) | 8.75% | 1865 |
| CoT | 2.08% | 1024 |
| CoT (Greedy) | 3.33% | 1024 |
| Soft Thinking | 0.00% | 1024 |
| SwiR (Ours) | 4.58% | 787 |

Table 21: Evaluation results of DeepSeek-R1-Distill-Llama-8B on the GSM8K benchmark under varying token budgets. Rows are *sorted by generation length in descending order*.

| Method | Accuracy (%) | Generation Length |
|---|---|---|
| Soft Thinking | 85.90% | 2953 |
| Soft Thinking | 84.84% | 2516 |
| CoT (Greedy) | 85.82% | 2393 |
| Soft Thinking | 83.62% | 2266 |
| Soft Thinking | 77.41% | 1741 |
| CoT (Greedy) | 86.05% | 1642 |
| CoT | 89.46% | 1588 |
| SwiR (Ours) | 90.07% | 1565 |
| CoT | 88.10% | 1554 |
| CoT | 88.02% | 1491 |
| CoT (Greedy) | 85.82% | 1421 |
| CoT | 86.13% | 1404 |
| SwiR (Ours) | 88.55% | 1349 |
| SwiR (Ours) | 89.46% | 1312 |
| CoT (Greedy) | 85.44% | 1307 |
| Soft Thinking | 70.36% | 1279 |
| SwiR (Ours) | 87.26% | 1217 |
| CoT | 78.62% | 1191 |
| CoT (Greedy) | 79.53% | 1092 |
| SwiR (Ours) | 86.43% | 1071 |
| Soft Thinking | 49.58% | 885 |
| CoT | 53.22% | 883 |
| CoT (Greedy) | 57.39% | 839 |
| SwiR (Ours) | 83.62% | 775 |
| CoT | 27.37% | 509 |
| CoT (Greedy) | 28.81% | 508 |
| Soft Thinking | 28.35% | 508 |
| SwiR (Ours) | 70.96% | 270 |
| CoT | 5.69% | 256 |
| CoT (Greedy) | 6.52% | 256 |
| Soft Thinking | 5.46% | 256 |

Table 22: Evaluation results of DeepSeek-R1-Distill-Llama-8B on the Math500 benchmark under varying token budgets. Rows are *sorted by generation length in descending order*.

| Method | Accuracy (%) | | Generation Length | |
|---|---|---|---|---|
| Soft Thinking | | 83.80% | | 4718 |
| CoT (Greedy) | | 84.80% | | 4110 |
| Soft Thinking | | 82.60% | | 4085 |
| SwiR (Ours) | | 92.00% | | 3837 |
| CoT | | 91.40% | | 3792 |
| CoT | | 89.80% | | 3572 |
| Soft Thinking | | 75.40% | | 3204 |
| CoT (Greedy) | | 83.20% | | 3203 |
| SwiR (Ours) | | 89.80% | | 3046 |
| SwiR (Ours) | | 88.00% | | 2931 |
| CoT | | 85.20% | | 2828 |
| SwiR (Ours) | | 86.40% | | 2722 |
| CoT (Greedy) | | 79.60% | | 2622 |
| SwiR (Ours) | | 86.00% | | 2462 |
| Soft Thinking | | 64.40% | | 2396 |
| CoT | | 71.00% | | 2133 |
| CoT (Greedy) | | 71.20% | | 2081 |
| SwiR (Ours) | | 79.20% | | 1953 |
| Soft Thinking | | 42.60% | | 1605 |
| CoT | | 47.60% | | 1539 |
| CoT (Greedy) | | 52.60% | | 1500 |
| SwiR (Ours) | | 68.40% | | 1116 |
| CoT | | 12.20% | | 512 |
| Soft Thinking | | 10.20% | | 511 |
| CoT (Greedy) | | 11.80% | | 511 |
| SwiR (Ours) | | 57.40% | | 453 |

Table 23: Evaluation results of DeepSeek-R1-Distill-Llama-8B on the GPQA Diamond benchmark under varying token budgets. Rows are *sorted by generation length in descending order*.

| Method | Accuracy (%) | | Generation Length | |
|---|---|---|---|---|
| Soft Thinking | | 33.33% | | 8593 |
| CoT | | 46.46% | | 7591 |
| Soft Thinking | | 30.81% | | 7507 |
| SwiR (Ours) | | 47.98% | | 7458 |
| CoT | | 46.46% | | 7236 |
| SwiR (Ours) | | 45.45% | | 6635 |
| SwiR (Ours) | | 44.44% | | 6038 |
| CoT (Greedy) | | 31.82% | | 5854 |
| SwiR (Ours) | | 41.92% | | 5406 |
| CoT (Greedy) | | 27.78% | | 5230 |
| CoT | | 31.31% | | 4943 |
| Soft Thinking | | 16.67% | | 4510 |
| SwiR (Ours) | | 39.90% | | 4388 |
| CoT (Greedy) | | 23.73% | | 4301 |
| SwiR (Ours) | | 41.41% | | 3292 |
| CoT (Greedy) | | 11.62% | | 2625 |
| Soft Thinking | | 6.57% | | 2350 |
| CoT | | 9.60% | | 2314 |
| SwiR (Ours) | | 25.76% | | 1840 |
| CoT | | 2.53% | | 1595 |
| CoT (Greedy) | | 3.54% | | 1469 |
| Soft Thinking | | 3.03% | | 1381 |
| CoT (Greedy) | | 0.50% | | 918 |
| CoT | | 1.01% | | 855 |
| Soft Thinking | | 1.01% | | 785 |
| SwiR (Ours) | | 29.80% | | 673 |
| CoT | | 0.00% | | 512 |
| CoT (Greedy) | | 0.00% | | 512 |
| Soft Thinking | | 0.00% | | 511 |

Table 24: Evaluation results of DeepSeek-R1-Distill-Llama-8B on the AIME2024 benchmark under varying token budgets. Rows are *sorted by generation length in descending order*.

| Method | Accuracy (%) | Generation Length |
|---|---|---|
| SwiR (Ours) | 45.00% | 8179 |
| CoT | 43.75% | 8145 |
| CoT (Greedy) | 30.00% | 7840 |
| SwiR (Ours) | 44.17% | 7555 |
| CoT | 41.25% | 7330 |
| CoT (Greedy) | 26.66% | 7107 |
| Soft Thinking | 34.17% | 6956 |
| SwiR (Ours) | 40.42% | 6803 |
| SwiR (Ours) | 41.25% | 6645 |
| SwiR (Ours) | 39.58% | 5876 |
| Soft Thinking | 32.08% | 5871 |
| CoT | 28.33% | 5086 |
| CoT (Greedy) | 16.67% | 4952 |
| Soft Thinking | 27.50% | 4860 |
| SwiR (Ours) | 32.50% | 4757 |
| CoT (Greedy) | 6.67% | 3795 |
| Soft Thinking | 11.67% | 3784 |
| CoT | 14.58% | 3515 |
| SwiR (Ours) | 23.33% | 3419 |
| SwiR (Ours) | 8.75% | 2103 |
| CoT | 1.67% | 2048 |
| CoT (Greedy) | 3.33% | 2048 |
| Soft Thinking | 3.33% | 2045 |

Table 25: Evaluation results of DeepSeek-R1-Distill-Llama-8B on the AIME2025 benchmark under varying token budgets. Rows are *sorted by generation length in descending order*.

| Method | Accuracy (%) | Generation Length |
|---|---|---|
| Soft Thinking | 20.42% | 8448 |
| SwiR (Ours) | 31.25% | 6827 |
| Soft Thinking | 19.17% | 6824 |
| CoT | 26.25% | 6583 |
| SwiR (Ours) | 30.83% | 6419 |
| CoT (Greedy) | 30.00% | 6293 |
| SwiR (Ours) | 30.00% | 6230 |
| CoT | 25.00% | 5724 |
| SwiR (Ours) | 29.17% | 5721 |
| SwiR (Ours) | 26.67% | 5229 |
| CoT (Greedy) | 26.67% | 4967 |
| CoT (Greedy) | 23.33% | 4370 |
| Soft Thinking | 14.17% | 4170 |
| CoT | 19.58% | 4085 |
| SwiR (Ours) | 21.25% | 4035 |
| CoT | 14.17% | 3197 |
| Soft Thinking | 10.00% | 3023 |
| SwiR (Ours) | 16.67% | 2970 |
| CoT (Greedy) | 16.67% | 2862 |
| CoT | 3.33% | 2048 |
| Soft Thinking | 0.00% | 2048 |
| CoT (Greedy) | 3.33% | 1904 |
| SwiR (Ours) | 7.08% | 1777 |

