# OpenReview forum: "SwiReasoning: Switch-Thinking in Latent and Explicit for Pareto-Superior Reasoning LLMs"
_ICLR.cc/2026/Conference — ICLR 2026 Poster_

### Official Review · Reviewer_JcUf · 2025-10-31

**Soundness:** 3
**Presentation:** 3
**Contribution:** 2
**Rating:** 6
**Confidence:** 3

**Summary:**

This paper presents **SWiReasoning**, a training-free framework that dynamically switches between **explicit (token-based)** and **latent (hidden-space)** reasoning based on entropy-derived confidence trends. A switch-count control prevents overthinking, improving both accuracy and token efficiency. Across multiple math and STEM reasoning benchmarks (GSM8K, Math500, AIME24/25, GPQA) and three reasoning LLMs (Qwen3-8B, Qwen3-1.7B, DeepSeek-R1), the method yields +1.5–2.8% Pass@1 improvement and up to +80% token efficiency gains without retraining.

**Strengths:**

- **Practical and training-free**: Improves reasoning efficiency entirely at inference time.
- **Comprehensive experiments**: Consistent gains across models and reasoning benchmarks.
- **Well-analyzed switching mechanism**: Includes ablations on window size, confidence, and signal mixing.
- **Clear motivation**: Tackles overthinking and token inefficiency in CoT reasoning.

**Weaknesses:**

- **Moderate novelty**: Combines existing ideas (latent reasoning + token control) with minor algorithmic extensions.
- **Pipeline complexity**: The overall system involves multiple heuristic components (entropy tracking, switching windows, signal mixing), making the pipeline heavy and less theoretically elegant.
- **System-oriented nature**: The contribution is more of a practical inference-time system design than a fundamentally new reasoning principle.
- **Marginal improvement**: Accuracy gains are modest given the added complexity.

**Questions:**

1. How robust is the entropy-based switching to unseen data or noisy prompts?
2. Could the switching mechanism be learned rather than manually tuned?
3. How does this compare to dynamic decoding methods such as COCONUT or Long⊗Short?

---

> ### Author Response · Authors · 2025-11-19
> **Responses to Reviewer JcUf [1/2]**
>
> Dear Reviewer JcUf,
>
> We thank you for highlighting our method’s practicality, comprehensive experiments with analyses, and clear motivation. We appreciate the constructive comments and have addressed all your concerns as follows.
>
> ---
>
> **Q1. Robustness to unseen data and noisy prompts**
>
> We agree that robustness is important. To address this, we conducted **new experiments in Appendix C.3 on code generation tasks**, which feature significantly different data types and prompt structures compared to math/STEM tasks.
>
> | Method         | HumanEval | LeetCode-Easy | LeetCode-Medium | LeetCode-Hard | MBPP | LiveCodeBench | Average    |
> |----------------|-----------|-----------------------|-------------------------|-----------------------|-----------------|-----------------|-----------|
> | CoT            | 92.68%     | 57.78%                 | 68.13%                   | 43.18%                 | 94.16% | 62.01%      | 77.07%   |
> | CoT (Greedy)   | 93.90%     | 64.44%                 | 58.24%                   | 47.73%              | 91.44% |   50.18% | 73.19%  |
> | Soft Thinking  | 92.07%     | 55.56%                | 61.54%                   | 38.64%               | 94.16% |  56.99% | 74.42%  |
> | **SwiR (Ours)**| **95.73%** | **64.44%**  | **69.23%**  | **61.36%**  | **95.33%** | **63.44%**  | **80.15%**  |
> | $\Delta$ | +3.05% | +6.66% | +1.10% | +18.18% | +1.17% | +1.43% | +3.08% |
>
>
> Specifically, we use Qwen3-8B and report pass@1 accuracy on HumanEval [1], LeetCode-Contest [2], MBPP [3] and LiveCodeBench [4] benchmarks. As shown in Tab. 6 of our updated version, under no token budget constraint, SwiReasoning improves accuracy by **+3.08%** on average over standard CoT (the reported average is the simple mean of accuracy over the full LeetCode-Contest and accuracies over three other benchmarks). The largest gains (**+18.18%**) are observed on the hard-level subset, indicating that SwiReasoning is most helpful for problems that require stronger reasoning capabilities.
>
> These results demonstrate that SwiReasoning effectively generalizes to different data types and prompt styles. Furthermore, as a training-free framework, SwiReasoning operates on the base model’s intrinsic representations. Our results suggest that it successfully preserves the base model’s inherent robustness rather than introducing new fragilities.
>
> ---
>
> **Q2. About learning the switch mechanism**
>
> This is a very insightful suggestion. In this work, we focused on a **training-free** design
> to maximize applicability and avoid additional training costs. Despite being out of scope of this work, we mentioned the same point in our conclusion that SwiReasoning with a training-based strategy, *e.g.*, reinforcement learning, is a promising direction for future work.
>
> ---
>
> **Q3. Compared to dynamic decoding methods**
>
>
> Thank you for highlighting these inspiring works. We have already had a discussion on Coconut in Sec. 2, and have added a new brief discussion on Long $\otimes$ Short in the **updated Appendix B.4**.
>
> The key distinction lies in the **cost**:
>
> * Methods like COCONUT and Long $\otimes$ Short require additional training. For example, SFT for COCONUT and RL for Long $\otimes$ Short alter the model’s generation behavior. While effective, this incurs non-negligible training costs and requires access to model weights and gradients.
>
> * Our work differs in principle: it is training-free and intervenes in real time to optimize the reasoning path of a frozen model. This makes our method orthogonal to learning-based methods like Coconut and Long $\otimes$ Short.
>
> ---

---

> ### Author Response · Authors · 2025-11-19
> **Responses to Reviewer JcUf [2/2]**
>
> **W1. Regarding novelty**
>
> We appreciate your perspective. While our framework leverages latent reasoning and token control, the novelty lies in the synergistic, bottom-up design that addresses the specific limitations of each component when used in isolation:
>
> 1. **Why Entropy-Based Switching?** Remaining in a single mode throughout reasoning is inherently suboptimal: explicit thinking provides readability but may discard useful information beyond chosen tokens, while latent thinking preserves richer signals but can drift into noise and reduce accuracy. Our key insight is that reasoning should switch modes based on confidence. Latent reasoning enables exploration across multiple potential continuations when confidence is low, and explicit reasoning encourages convergence when confidence is high, striking a balance that supports broad exploration while maintaining accuracy.
>
> 2. **Why Token Control?** Even with confidence-aware switching, reasoning LLMs may still overthink. Therefore, we place a bound on the total number of switches. Our key insight is that each switch naturally marks the end of a thinking block where partial reasoning trajectories have been consolidated, which may already contain sufficient evidence for arriving at a reasonable solution. Under limited budgets, generating answers at these natural checkpoints can make use of partial reasoning trajectories, offering a chance to obtain correct predictions earlier without consuming additional tokens.
>
> 3. **The Synergy**: The combination is not merely additive. It is the switching mechanism that creates these natural checkpoints that are suitable for the token controller to exploit.
>
> ---
>
> **W2. Pipeline complexity**
>
> We understand the concern regarding complexity. We wish to clarify that the “complexity” is largely internal to the framework’s design, not user-facing. Most components are **fixed** across all benchmarks and models, requiring no per-task tuning. For instance, the mentioned entropy tracking with an asymmetric switch window size of 512 and signal mixing coefficient $\beta_0$=0.7. Further details are also summarized in Appendix B.3.
>
> Our design philosophy is to provide users with an appropriate level of flexibility to trade off between latency and accuracy by exposing minimal hyperparameters, which we believe are necessary for a practical, training-free reasoning framework. In practice, users are provided with the flexibility to adjust $\alpha_0$ based on task difficulty, and optionally, for further efficiency, $C_{max}$ based on budget constraints. All other designs and hyperparameters are fixed inside the framework. This flexibility allows users to adapt SwiReasoning to different task difficulties and efficiency requirements. As you noted in our ablations, Tab. 2 and 3 show that adjustable hyperparameters are not overly sensitive within a broad plateau, and the chosen values of the fixed ones perform well across various scenarios.
>
> ---
>
> **W3. System-oriented nature**
>
> We acknowledge the system-oriented contributions of our work. We do not view it as a concern but more as diverse research paths. We agree that inventing a “fundamentally new reasoning principle” could be valuable, but practical frameworks compatible with existing systems, like ours, are equally valuable. Looking back at the history of AI, these two research paths have often evolved in a spiral pattern, complementing each other. As you noted, SwiReasoning offers **immediate practical value** to existing reasoning systems and could complement other fundamental research.
>
> ---
>
> **W4. Accuracy improvement**
>
> We would like to respectfully clarify that our gains are **substantial**, particularly considering their training-free nature. As shown in Table 1, our work achieves +2.17% average improvement across various models and benchmarks, and on most challenging benchmarks (AIME24/25), our gains are as high as +5.00%. Additionally, new experiments across code domains suggest that SwiReasoning significantly improves accuracy, particularly on hard-level LeetCode problems, with a +18.18% improvement. All improvements are gained without the need for new data or further training.
>
> ---
>
> Thank you again for your thoughtful suggestions, which have motivated us to conduct additional experiments that have strengthened the paper. We’d appreciate your re-evaluation based on the updated version. Please do not hesitate to let us know if you have any further comments. We will be glad to discuss them.
>
> ---
>
> **References**
> [1] “Evaluating Large Language Models Trained on Code” by OpenAI, 2021.
> [2] “DeepSeek-Coder: When the Large Language Model Meets Programming - The Rise of Code Intelligence” by DeepSeek, 2024.
> [3] “Program Synthesis with Large Language Models” by Austin et al. ArXiv’21.
> [4] “Livecodebench: Holistic and contamination free evaluation of large language models for code” by Jain et al. ArXiv’ 24.

---

> > ### Comment · Reviewer_JcUf · 2025-11-27
> >
> > Thank you for the rebuttal. The authors’ clarifications resolved my concerns.

---

> ### Author Response · Authors · 2025-11-27
>
> Thank you for reviewing the rebuttals and for your positive feedback! We are glad that our clarifications have resolved your concerns. We sincerely appreciate your suggestions, which have helped improve this work.

---

### Official Review · Reviewer_BFnL · 2025-11-03

**Soundness:** 2
**Presentation:** 2
**Contribution:** 2
**Rating:** 4
**Confidence:** 5

**Summary:**

The paper proposes SWIREASONING, a training-free framework that dynamically switches between explicit and latent reasoning based on entropy-driven confidence signals in the next-token distribution. The key idea is that staying purely in latent space can diffuse probability mass and cause slow or noisy convergence, so the model alternates modes to balance exploration and exploitation. A second mechanism caps the number of thinking-block switches to reduce “overthinking” and save tokens. The authors conduct some experiments and show the performance.

**Strengths:**

1.	The paper proposes a method that flexibly combines latent reasoning with explicit reasoning, effectively reusing intermediate embeddings as inputs to enable denser thinking steps.
2.	The authors provide experiments on several standard reasoning benchmarks to demonstrate the applicability of the approach.

**Weaknesses:**

1.	The overall performance gains are relatively small, and the efficiency story is not fully convincing — a clearer, more direct metric is needed, and Appendix Table 8 suggests that long generations are still required to reach good accuracy.
2.	The paper should further explain why switching between latent and explicit modes improves performance, and clarify whether (and how) the latent trajectories can be decoded or inspected in text for interpretability.
3.	The experimental scope is limited; results on larger models and on other task types (e.g., coding), where latent reasoning may be less naturally applicable, would make the empirical claims more convincing.

**Questions:**

Please chek the weakness

---

> ### Author Response · Authors · 2025-11-19
> **Responses to Reviewer BFnL [1/3]**
>
> Dear Reviewer BFnL,
>
> We thank you for acknowledging the flexibility of our training-free framework and the breadth of our empirical studies. We appreciate the constructive suggestions and have addressed all your concerns as follows.
>
> ---
>
> **W3. Experiments on larger models and more task types**
>
> We agree that demonstrating scalability on larger models and generalization on more task types is important. To address this, we conducted **two sets of new experiments** (detailed in **Appendix C.2** and **C.3**):
>
> | Method         | GSM8K        | MATH 500     | GPQA Diamond | AIME 2024    | AIME 2025    | Average        |
> |---------------|-------------:|-------------:|-------------:|-------------:|-------------:|---------------:|
> | CoT           | 95.83        | 97.40        | 66.16        | 80.42        | 72.08        | 82.38  |
> | CoT (Greedy)  | 95.91        | 97.20        | 69.70        | 80.00        | 73.33        | 83.23  |
> | Soft Thinking | 95.75        | 97.40        | 67.17        | 74.58        | 66.25        | 80.23  |
> | SwiR (Ours)   | **96.21** | **98.40** | **70.20** | **82.92** | **73.75** | **84.30** |
> | $\Delta$ | +0.38 | +1.00 | +4.04 | +2.50 | +1.67 | +1.92 |
>
>
> * **On the large-scale Qwen3-32B model**: We extended our evaluation to the 32B scale. As shown in **Tab. 5** of the updated version, SwiReasoning consistently improves performance, achieving an average gain of **+1.92%** over standard CoT under no token budget constraint. The gains are most notable on more difficult benchmarks such as GPQA Diamond (**+4.04%**). These results confirm that SwiReasoning **scales effectively to larger models**.
>
> | Method         | HumanEval | LeetCode-Easy | LeetCode-Medium | LeetCode-Hard | MBPP | LiveCodeBench | 2WikiMultihopQA | CommonsenseQA | Average    |
> |----------------|-----------|-----------------------|-------------------------|-----------------------|-----------------|-----------------|-----------------|---------------|-----------|
> | CoT            | 92.68%     | 57.78%                 | 68.13%                   | 43.18%                 | 94.16% | 62.01% | 79.00%           | 83.95%         | 78.54%   |
> | CoT (Greedy)   | 93.90%     | 64.44%                 | 58.24%                   | 47.73%              | 91.44% |   50.18% | 79.50%           | 83.95%         | 76.03%  |
> | Soft Thinking  | 92.07%     | 55.56%                | 61.54%                   | 38.64%               | 94.16% |  56.99% | 79.00%           | 83.70%         | 76.73%  |
> | **SwiR (Ours)**| **95.73%** | **64.44%**  | **69.23%**  | **61.36%**  | **95.33%** | **63.44%** | **81.50%**  | **85.34%**  | **81.24%**  |
> | $\Delta$ | +3.05% | +6.66% | +1.10% | +18.18% | +1.17% | +1.43% | +2.50% | +1.39% | +2.70% |
>
> * **On coding tasks**: In addition to math and STEM (including physics, chemistry, biology, and computer science) domains, we evaluate Qwen3-8B and report pass@1 accuracy on HumanEval [1], LeetCode-Contest [2], MBPP [4] and LiveCodeBench [5] benchmarks. As shown in Tab. 6 of our updated version, under no token budget constraint, SwiReasoning improves accuracy by **+3.08%** on average over standard CoT (the reported average is the simple mean of accuracy over the full LeetCode-Contest and accuracies over three other coding benchmarks). The largest gains (**+18.18%**) are observed on the hard-level subset, indicating that SwiReasoning is most helpful for problems that require stronger reasoning capabilities.
> * **On multi-hop QA** tasks, which require retrieving and connecting multiple disparate facts, SwiReasoning outperforms CoT by +2.50%. This suggests that the exploration capability of latent reasoning is effective in navigating complex reasoning paths.
> * **On commonsense reasoning** tasks, SwiReasoning surpasses CoT by +1.39%. This demonstrates the robustness of SwiReasoning in general knowledge scenarios.
>
> These results confirm that SwiReasoning **generalizes to broader domains with consistent accuracy gains**.
>
> ---

---

> ### Author Response · Authors · 2025-11-19
> **Responses to Reviewer BFnL [2/3]**
>
> **W1. Gains and efficiency metrics**
>
> * **Regarding gains**: we would like to respectfully clarify that our gains are **substantial**, especially given the **training-free** nature of our method. As shown in Table 1, our work achieves +2.17% average improvement across various models and benchmarks, and on most challenging benchmarks (AIME24/25), our gains are as high as +5.00%. Additionally, new experiments across code domains suggest that SwiReasoning significantly improves accuracy, particularly on hard-level LeetCode problems, with a +18.18% improvement. All improvements are gained without the need for new data or further training.
>
> | Latency (s) | Acc=95% | Acc=90% | Acc=80% |
> |--------------|-----------------|-----------------------------|-------------------------------|
> | CoT          |  99.9 | 88.24 | 71.54 |
> | CoT (Greedy) | 103.07 | 86.03 |  69.66 |
> | Soft Thinking | 100.27 | 88.35 | 71.25 |
> | SwiR (Ours)     | **84.89**  | **64.82** | **32.99** |
> | $\Delta$ | 1.18x faster | 1.36x faster | 2.17x faster |
>
> | TFLOPs | Acc=95% | Acc=90% | Acc=80% |
> |--------------|-----------------|-----------------------------|-------------------------------|
> | CoT          | 78.69  | 70.13  | 57.04 |
> | CoT (Greedy)    | 82.40 | 69.25 | 55.75 |
> | Soft Thinking  | 79.08  | 70.24 | 57.03 |
> | SwiR (Ours)     | **68.30** | **51.72** | **26.22** |
> | $\Delta$ | 1.15x fewer | 1.36x fewer | 2.18x fewer |
>
> * **Regarding efficiency metrics**: As you suggested, to provide efficiency measurements in more metrics, we have added new **wall-clock latency** and **TFLOPs** experiments in **Appendix C.4**. **Fig. 7** in the updated version demonstrates that SwiReasoning consistently reaches a given accuracy with lower latency and fewer TFLOPs, and the advantage grows as the budget becomes tighter. Specifically, at 90% pass@1 accuracy, SwiReasoning uses 1.36× fewer TFLOPs and is 1.36× faster in wall-clock time than CoT. At 80% accuracy, it uses 2.18× fewer TFLOPs and is 2.17× faster. These results confirm that SwiReasoning improves efficiency at fixed accuracies, especially under small budgets.
>
> * **Regarding Appendix Tab. 8** (Tab. 14 in the updated version): Thank you for examining the supplementary data. The results here illustrate our twofold contributions:
>
>   1. **Higher Performance Ceiling**: Under unlimited budgets, we achieve higher peak accuracy (79.17% vs 75.83% for CoT), which is significant as accuracy saturates as generation length continues to grow.
>
>   2. **Superior Pareto Efficiency**: In limited budget settings (the majority of data points in the mentioned table), SwiReasoning consistently matches or exceeds baseline performance while using much fewer tokens.
>
>   Overall, SwiReasoning improves the performance ceiling when there is no budget constraint and improves efficiency when the budget is limited.
>
> ---

---

> ### Author Response · Authors · 2025-11-19
> **Responses to Reviewer BFnL [3/3]**
>
> **W2. Why does it work & inspecting latent trajectories**
>
> Thank you for your perspective.
>
> 1. **Why does it work?** Remaining in a single mode throughout reasoning is inherently suboptimal: explicit thinking provides readability but may discard useful information beyond chosen tokens, while latent thinking preserves richer signals but can drift into noise and reduce accuracy. Our key insight is that reasoning should switch modes based on confidence. Latent reasoning enables exploration across multiple potential continuations when confidence is low, and explicit reasoning encourages convergence when confidence is high, striking a balance that supports broad exploration while maintaining accuracy.
>
> 2. **Inspecting latent trajectories**. We wish to clarify that the non-decodability of latent trajectories is an inherent trade-off of this class of methods (originating from works like Coconut [3]), rather than a specific limitation of SwiReasoning. Latent states are weighted averages of the vocabulary (Eq. 1) and cannot be losslessly decoded into text.
>
>    * **Improved Interpretability**: However, SwiReasoning offers **better** interpretability than pure latent methods like Coconut or Soft Thinking, because it switches between latent and explicit text. The explicit segments provide readable checkpoints throughout the reasoning process.
>
>    * **A promising direction**: We agree that analyzing the interpretability properties of latent states (*e.g.*, fetching a set of top-p token candidates in latent space, calculating their semantic distances, and analyzing the overall information) is a promising direction for the field, though it is distinct from the scope of accuracy-efficiency improvement of this work.
>
> ---
>
> Thank you again for your thoughtful suggestions, which have motivated us to conduct additional experiments that have strengthened the paper. We’d appreciate your re-evaluation based on the updated version. Please do not hesitate to let us know if you have any further comments. We will be glad to discuss them.
>
> ---
>
> **References**
> [1] “Evaluating Large Language Models Trained on Code” by OpenAI, 2021.
> [2] “DeepSeek-Coder: When the Large Language Model Meets Programming - The Rise of Code Intelligence” by DeepSeek, 2024.
> [3] “Training Large Language Models to Reason in a Continuous Latent Space” by Hao et al. COLM 2025.
> [4] “Program Synthesis with Large Language Models” by Austin et al. ArXiv’21.
> [5] “Livecodebench: Holistic and contamination free evaluation of large language models for code” by Jain et al. ArXiv’ 24.

---

### Official Review · Reviewer_eVQp · 2025-11-03

**Soundness:** 3
**Presentation:** 3
**Contribution:** 3
**Rating:** 6
**Confidence:** 4

**Summary:**

This paper introduces SwiReasoning, a training-free framework for improving reasoning in large language models by dynamically switching between explicit chain-of-thought (CoT) and latent reasoning modes. The core innovation lies in using entropy trends from next-token distributions as block-wise confidence signals to guide mode transitions: when confidence increases (entropy drops), the system switches to explicit reasoning to consolidate progress; when uncertainty persists (entropy rises), it switches to latent reasoning for exploration. Additionally, a switch count control mechanism limits the maximum number of mode transitions and enables early answer generation at natural checkpoints, suppressing overthinking and improving token efficiency.

**Strengths:**

1. The insight that reasoning should adaptively switch based on confidence is intuitive yet underexplored in training-free settings, thereby making the idea and approach of the paper novel.

2. The evaluation is comprehensive and spans multiple model families, scales (1.7B–8B parameters), training paradigms, and five diverse benchmarks. The consistency of improvements across all settings (13 out of 15 evaluations achieve highest token efficiency) provides strong empirical support.

3. The methodology is well-structured with intuitive explanations, formal algorithmic descriptions (Algorithm 1), and effective visualizations (Figures 1–5). The paper is well written.

**Weaknesses:**

Enumerating a few weaknesses here

1. While the convergence and termination triggers are intuitive, the specific thresholds (½Cmax for convergence, Cmax for termination) appear arbitrary. Why these specific fractions? How sensitive is performance to these choices? The paper doesn't ablate these design decisions.

2. Limited failure case analysis and the modes of failure. It would be good to add the cases or categories of questions where the proposed approach fails.

3. The results reported for soft thinking approach appear to be lower than standard CoT, which is contrary to the claims in the original Soft-thinking paper. This raises concerns about implementation correctness and whether comparisons are on equal footing.

**Questions:**

1. How is entropy computed during latent steps when no explicit token is selected? Is it from the softmax over the last hidden state projected through the LM head?

2. In ablation Table 3, window size 512 is optimal. Given typical generation lengths (e.g., ~2000 tokens for GSM8K), does this mean most problems only switch 2–4 times? If so, is this really "dynamic" switching, or just a staged approach (latent -> explicit -> latent -> explicit) Can you show histograms of actual switch counts across problem difficulties?

3. The paper claims "Pareto-superior" results. However, in some cases (e.g., AIME24 in Fig. 4), CoT appears to match or exceed SwiReasoning's accuracy at similar token budgets. Can you clarify where strict Pareto dominance holds vs. where there are trade-offs?

---

> ### Author Response · Authors · 2025-11-21
> **Responses to Reviewer eVQp [1/2]**
>
> Dear Reviewer eVQp,
>
> We thank you for the positive assessment of our work's novelty, comprehensive evaluation, and clear presentation. We appreciate your insightful questions which have guided us to conduct new experiments that strengthened the paper. We addressed all your comments as follows.
>
> ---
>
> **W1. Ablation on convergence triggers vs. termination triggers**
>
> Thank you for this suggestion. As requested, we have added **a new ablation study in Appendix C.7**. To analyze the positions of convergence triggers w.r.t. termination ones, we conducted an ablation study using Qwen3-1.7B on the Math500 benchmark. We fixed the termination trigger at two representative budgets ($C_{\max}=8$ and $C_{\max}=20$) and varied the position of the convergence trigger relative to the termination step. Specifically, if $T$ denotes the switch count at termination (*i.e.*, $T=C_{\max}$), we tested setting the convergence trigger at $C \in \\{\frac{1}{4}T, \frac{1}{3}T, \frac{1}{2}T, \frac{2}{3}T, \frac{3}{4}T\\}$. Tab. 7 in the update version presents a clear trade-off between accuracy and token efficiency:
>
> | $C_{\max}$ | Metric            |$C=\tfrac{1}{4}T$ | $C=\tfrac{1}{3}T$ | $C=\tfrac{1}{2}T$ | $C=\tfrac{2}{3}T$ | $C=\tfrac{3}{4}T$ |
> |-------------|-------------------|---------------------|---------------------|---------------------|---------------------|---------------------|
> | 8           | Accuracy (%)      | 76.60               | 79.20               | 81.00               | 84.20               | 84.20               |
> | 8           | Generation Length | 1428                | 1817                | 2222                | 2743                | 2743                |
> | 20          | Accuracy (%)      | 84.00               | 85.80               | 87.80               | 88.20               | 88.80               |
> | 20          | Generation Length | 2788                | 3278                | 3597                | 4002                | 3943                |
>
> * **Early convergence** ($\frac{1}{4}T, \frac{1}{3}T$): triggering convergence too early significantly reduces generation length but leads to a notable drop in accuracy (*e.g.*, at $C_{\max}=20$, accuracy drops from $87.80\%$ to $84.00\%$ when moving from $\frac{1}{2}T$ to $\frac{1}{4}T$). This suggests that forcing the model to converge before sufficient exploration prevents it from finding correct solutions for harder problems.
>
> * **Late convergence** ($\frac{2}{3}T, \frac{3}{4}T$): delaying the convergence trigger yields marginal accuracy gains (*e.g.*, +$1.0\%$ from $\frac{1}{2}T$ to $\frac{3}{4}T$ at $C_{\max}=20$) but comes at the cost of increased token consumption.
>
> * **Balanced default** ($\frac{1}{2}T$): the default setting of $\frac{1}{2}C_{\max}$ effectively strikes a balance. It captures the majority of the accuracy while maintaining reasonable token efficiency. Overall, $C=\frac{1}{2}T$ serves as a sweet spot for general-purpose reasoning under budget constraints.
>
> ---
>
> **W2. Analysis of challenging scenarios**
>
> Thank you for your suggestion. We agree that challenging task analysis adds depth. While SwiReasoning demonstrates robust generalization across diverse domains, we identify a specific category of tasks, **spatial reasoning with rigid topological constraints**, where certain instances may be challenging for latent-based methods. A representative case is the “3D Surface Shortest Path” problem from the Math500 benchmark shown in Appendix C.6. In this problem, a gecko must find the shortest path on a 3D room’s surface (we highly recommend you check Appendix C.6 directly for better clarity).
>
> The reasoning trace reveals that SwiReasoning correctly identifies the high-level strategy: unfolding the 3D room into a 2D plane. However, it struggles to maintain the specific topological “only walk across the ceiling and the walls” constraint during latent transitions. We attribute this behavior to a **trade-off inherent in latent reasoning**:
>
> * **Smoothing vs. Precision**: Latent reasoning operates by smoothing probability distributions to enable diverse semantic exploration. While this is beneficial for logical reasoning, it may be detrimental for tasks with precise constraints, such as rigid geometric topology (*e.g.*, defining exactly which wall edge connects to which).
>
> * **Blurring Constraints**: During the latent mode, the strict “only walk across the ceiling and the walls” constraint is blurred due to the spreading of probability mass. As a result, when the model switches back to explicit mode, it may reason with an invalid path (*e.g.*, calculating a direct Euclidean distance through prohibited space), leading to an incorrect answer ($4\sqrt{13}$ vs. true $2\sqrt{113}$).
>
> Overall, this suggests that while SwiReasoning excels at semantic and logical exploration, a few tasks requiring rigid geometric constraint satisfaction, where fuzzy latent exploration may interfere with precise geometric execution, could still be challenging.
>
> ---

---

> ### Author Response · Authors · 2025-11-21
> **Responses to Reviewer eVQp [2/2]**
>
> **W3. Baseline performance**
>
> We thank you for noting this. We’d like to clarify that we are using the correct implementation, and this performance discrepancy is not observed in our work alone. A concurrent work [6] also reported a **similar observation independently**. For example, in Tab. 1 of their paper, they found that soft thinking underperforms standard CoT with sampling, with an average accuracy decrease of 2.56%- 6.37%.
>
> We want to highlight that soft thinking is a very inspiring and insightful work. For this performance discrepancy observed by our work and the concurrent work, we believe it may be largely because soft thinking was originally evaluated on relatively older model families. As reasoning LLMs have been evolving fast recently, their baseline CoT capabilities have improved significantly. Overall, we assure the reviewer that the comparisons are **fair**.
>
> ---
>
> **Q1. Entropy calculation**
>
> Thank you for asking. The entropy $H_t$ is computed from the full next-token distribution $p_t$ at every step, before a mode-specific action is taken.
>
> * In **explicit** mode, we compute $p_t$, calculate $H_t$, and then sample $x_t \sim p_t$ as the input of the next step.
> * In **latent** mode, we compute $p_t$, calculate $H_t$, and then use $p_t$ to calculate latent embedding $\tilde{e}_t$ with Eq. 1 as the input of the next step.
>
> Overall, the entropy is calculated **identically** in both modes.
>
> ---
>
> **Q2. Histograms of actual switch counts**
>
> | Dataset | Qwen3-1.7B | Qwen3-8B | Qwen3-32B | Average over models |
> | ------------ | ---------: | -------: | --------: | ----------------: |
> | GSM8K |       7.73 |     7.63 |      5.65 |              7.00 |
> | Math500|      19.42 |    18.74 |     15.44 |             17.87 |
> | GPQA Diamond |    35.61 |  33.84  | 25.67 |  31.70 |
> | AIME 2024    |      68.80  | 56.60 |   44.40  | 56.60
> | AIME 2025    |      58.87  |  67.80  |  57.93  |  61.53
>
> Thank you for your suggestion. To investigate the actual switching dynamics, we analyzed the distribution of switch counts across five benchmarks of varying difficulty in **Appendix C.5**. Figure 8 presents the box plots of switch counts for Qwen3-1.7B, Qwen3-8B, and Qwen3-32B. Besides, the table above shows the average switch counts across datasets and models.
>
> The switch count distribution demonstrates strong adaptability to problem difficulty. In general, the difficulty ranking of the five tested benchmarks is GSMK8K < Math500 < GPQA Diamond < AIME2024 $\approx$ AIME 2025. On the simplest GSM8K, the medians of switching range from 6 to 8. On benchmarks requiring complex reasoning such as AIME 2024 and AIME 2025, the switch counts increase substantially, with medians ranging from 44 to 69, and maximum counts exceeding 150+. These results show that SwiReasoning is not a static staged process but **dynamically adapts to problem difficulty**.
>
> ---
>
> **Q3. Pareto-superiority and Fig. 4**
>
> Thank you for this close reading. The "Pareto-superior" claim refers to the **overall frontier**, *e.g.*, Fig. 2. Even though considerable overall performance improvement is achieved, we explicitly acknowledged a few sub-optimal efficiency cases in Sec 4.3: “In contrast, on AIME24/25, the efficiency gaps are smaller, while the accuracy gains with unlimited budgets (in Tab. 1) are larger. This asymmetry is expected: the harder the problem is, the more difficult it is to predict a correct answer with unfinished reasoning trajectories.” The trade-offs can be summarized as:
> * On easier problems like GSM8K, our gains are primarily **efficiency-driven** (same accuracy, much faster).
> * On harder problems like AIME24/25, our gains are primarily **capability-driven** (higher accuracy ceiling).
>
> ---
>
> Thank you again for your thoughtful suggestions, which have motivated us to conduct additional experiments that have strengthened the paper. We’d appreciate your re-evaluation based on the updated version. Please do not hesitate to let us know if you have any further comments. We will be glad to discuss them.
>
> ---
> **References**
> [1] LLMs are Single-threaded Reasoners: Demystifying the Working Mechanism of Soft Thinking. ArXiv’25.

---

> > ### Comment · Reviewer_eVQp · 2025-11-27
> > **Officient Comment by Reviewer eVQp**
> >
> > Thanks to the authors for conducting further ablation studies. These results further make the proposed approach strong. I will increase the score.

---

> > > ### Author Response · Authors · 2025-11-27
> > >
> > > Thank you for reviewing and acknowledging the new results! We are glad that the additional experiments have addressed your concerns and appreciate your comments for helping strengthen this work.

---

### Official Review · Reviewer_dZdK · 2025-11-03

**Soundness:** 3
**Presentation:** 3
**Contribution:** 2
**Rating:** 4
**Confidence:** 4

**Summary:**

This paper proposes SwiReasoning, a training-free framework for LLM reasoning that dynamically switches between explicit chain-of-thought and latent reasoning modes. The switching is guided by block-wise confidence estimated from entropy trends in next-token distributions. The method also includes a switch count controller to suppress overthinking and improve token efficiency. Experiments on mathematical and STEM benchmarks show consistent accuracy improvements and token efficiency gains across different model families and sizes.

**Strengths:**

- The idea of dynamically switching between explicit and latent reasoning based on confidence signals is intuitive and addresses limitations of pure latent reasoning approaches.
- The experimental evaluation is comprehensive across multiple models and benchmarks, including various token budget settings and Pass@k analysis that shows the method overall improves the performance with better efficiency.
- The ablation studies analyze key design choices and provide insights into how different components contribute to performance.

**Weaknesses:**

- The method introduces many heuristic design choices and hyperparameters (window sizes W_E→L and W_L→E, signal mixing coefficients α₀ and β₀, switch count limits C_max, convergence/termination triggers), and it appears from the ablations that these are tuned on the test sets, which raises concerns about generalization and fair comparison.

- It seems unclear whether the latent reasoning mode genuinely enables the claimed "exploration" of multiple reasoning paths, since Soft-Thinking simply weights tokens by probability, and the pretrained LLM likely still concentrates probability mass on similar natural language tokens. It would be beneficial to provide some analysis of the probability distributions within the latent tokens or other evidence that meaningful exploration is occurring beyond what explicit sampling would achieve.

- There are quite a few citations that are wrong and appear to be LLM-hallucinated, even including the most important "Soft-Thinking" paper that this work is primarily built on top of - the citation in the draft is ```Bowen Zhang, Yanzhuo Li, Shuohang Wang, Yu Tu, Jason Wei, Ashish Vaswani, Shikun Liu, Chenlu Yu, Deli Chen, Hongyi Li, Zhihua Zhang, and Chen Liang. Soft thinking: Training-free
latent reasoning for large language models.``` while the actual one should be ```Zhen Zhang, Xuehai He, Weixiang Yan, Ao Shen, Chenyang Zhao, Shuohang Wang, Yelong Shen, Xin Eric Wang. Soft Thinking: Unlocking the Reasoning Potential of LLMs in Continuous Concept Space```.

**Questions:**

See weaknesses.

---

> ### Author Response · Authors · 2025-11-18
> **Responses to Reviewer dZdK [1/2]**
>
> Dear Reviewer dZdK,
>
> We thank you for the positive feedback on our intuitive idea, thorough experimental evaluation, and insightful ablation studies. We appreciate the constructive comments and have addressed all your concerns as follows.
>
> ---
>
> **W1. Design choices and hyperparameters**
>
> We agree that clarity regarding design choices is important. We would like to clarify that most of them are **fixed** across all benchmarks and models. These choices are largely internal to the framework’s design, not user-facing. For instance, asymmetric window sizes = 512, signal mixing coefficients $\beta_0$=0.7, and the convergence and termination triggers are described in Sec. 3. Further details are also summarized in Appendix B.3.
>
> Our design philosophy is to provide users with an appropriate level of **flexibility by exposing minimal hyperparameters**, which we believe are necessary for a practical, training-free reasoning framework. In practice, users are provided with the flexibility to adjust $\alpha_0$ based on task difficulty, and optionally, for further efficiency, $C_{max}$ based on budget constraints. All other designs and hyperparameters are fixed inside the framework. This flexibility allows users to adapt SwiReasoning to different task difficulties and efficiency requirements. As you noted in our ablations, Tables 2 and 3 show that adjustable hyperparameters are not overly sensitive within a broad plateau, and the chosen values of the fixed ones perform well across various scenarios.
>
> To your point on the fairness of comparison, all method hyperparameters for baselines follow the recommendations from their original papers. All generation settings (*e.g.*, temperature, top-k) and prompt templates adhere to the default settings from the original models. This ensures the fairness of  our comparisons. Our additional experiments on larger models in Appendix C.2 and broader domains in Appendix C.3 also provide **clear evidence** for “generalization and fair comparison” futher.
>
> ---
>
> **W2. Analyses of the probability distributions within the latent mode**
>
> Thank you for your suggestion. To **quantitatively validate** the claimed “exploration,” as you suggested, we have added **two sets of new experiments in Appendix C.1**. Specifically, to quantify exploration in both the latent and explicit modes, we analyze two complementary statistics using the Qwen3-8B model on Math500, as visualized in **Fig. 6** of our updated version.
>
> | Mode             | Average Entropy |
> |------------------|--------------------------|
> | Latent Mode      | 0.444                    |
> | Explicit Mode    | 0.213                    |
> | Ratio (Latent / Explicit) | **2.08x**       |
>
> 1. **Shannon Entropy**: To measure the differences in the concentration level of the next-token distributions between latent and explicit modes, we calculate the average Shannon entropy across steps for each sample. Fig. 6(a) shows that the entropy in latent mode is **2.08x** higher than in explicit mode on average (0.444 vs. 0.213), indicating that the probability mass is spread across more candidate tokens in latent mode rather than collapsing.
>
> | Mode             | Average Similarity  |
> |------------------|---------------------------------------------|
> | Latent Mode      | 0.118                                       |
> | Explicit Mode    | 0.185                                       |
> | Ratio (Latent / Explicit) | **0.638x**                      |
>
> 2. **Semantic Diversity**: Entropy alone does not guarantee semantic diversity, as the semantic distances of different token candidates can be close, even when the probability mass is spread across them. To measure the differences in the semantic similarity among token candidates between latent and explicit modes, we calculate the average pair-wise cosine similarity of top-5 token candidates over steps by using their embeddings for each sample. Fig. 6(b) shows that the semantic similarity in latent mode is **63.8%** of that in explicit mode on average (0.118 vs. 0.185), showing that token candidates considered under latent modes are more semantically diverse.
>
> Taken together, these results provide quantitative evidence that latent modes enable **exploration with less concentrated token distributions and more semantically diverse candidates**.
>
> ---

---

> ### Author Response · Authors · 2025-11-18
> **Responses to Reviewer dZdK [2/2]**
>
> **W3. Inaccurate BibTeX entries appeared in 2 out of 70 references**
>
> Thank you for catching this glitch. We confirm that every citation in the paper corresponds to existing literature, and we did **not** use LLMs to generate non-existent references. This is a **formatting corruption of real citations**, not a hallucination of non-existent ones.
>
> By tracing the Overleaf revision history, we confirmed that the issue occurred when we were reformatting the BibTeX entry styles. The original BibTeX entries were accurate, but they used different styles, so we attempted to unify their formats automatically, which inadvertently caused the corruption. The **key evidence**, as you noted, is that the **method names in titles** “Soft Thinking” and “LLMs are single-threaded reasoners”  for the two affected entries **remain entirely accurate and map to existing literature**. This confirms we initially cited the correct existing literature, and the issue was a reformatting corruption rather than the generation of non-existent references.
>
> We have manually verified every reference in the paper and found that 2 out of 70 references were affected. They correspond to truly existing literature:
> * Soft thinking: Unlocking the reasoning potential of LLMs in continuous concept space (arXiv:2505.15778),
> * LLMs are single-threaded reasoners: Demystifying the working mechanism of soft thinking (arXiv:2508.03440).
>
> We confirmed that the other 68 references are accurate. We have fixed this issue in the updated version and would like to express our sincere apologies to the authors of these two papers. We want to assure the reviewer that every citation in the paper can be traced back to existing literature, and this formatting glitch **does not affect the central contributions, experimental results, or conclusions of the work**.
>
> ---
>
> Thank you again for your thoughtful suggestions, which have motivated us to conduct additional experiments that have strengthened the paper. We’d appreciate your re-evaluation based on the updated version. Please do not hesitate to let us know if you have any further comments. We will be glad to discuss them.

---

### Official Review · Reviewer_hrDS · 2025-11-04

**Soundness:** 3
**Presentation:** 4
**Contribution:** 2
**Rating:** 2
**Confidence:** 4

**Summary:**

Summary
This paper introduces SWIREASONING, a training-free reasoning approach that integrates latent and explicit chain-of-thought (CoT) modes. Instead of committing to token-level reasoning at every step, the method forms latent reasoning states via weighted token embeddings, allowing the model to retain uncertainty and explore multiple hidden trajectories.
A block-wise entropy reference is used as a switching criterion: when the entropy decreases (confidence rises), the model switches from latent to explicit reasoning immediately; when entropy increases, switching from explicit back to latent requires a dwell window to prevent oscillation. To signal mode transitions more explicitly, the method injects fused embeddings of  < think > / </ think > tokens during switch steps.
Additionally, switch counters and termination triggers are employed to avoid unbounded “over-thinking” and encourage timely final answer production.
Experiments show that, particularly for smaller models (≤8B) and token-budget-constrained settings, SWIREASONING delivers higher reasoning accuracy and efficiency compared to standard CoT and soft-thinking baselines. Ablation studies on signal mixing and dwell window length further support the design choices.

**Strengths:**

1. The approach does not require model updates and can be applied directly at inference time, making it practical for deployment on existing models.
2.The entropy based switching between latent and explicit reasoning, combined with a dwell window to avoid oscillation, is conceptually simple and computationally lightweight.
3. The method shows consistent gains over standard CoT on math reasoning benchmarks, particularly for ≤8B models under constrained token budgets, with ablation studies supporting key design choices.

**Weaknesses:**

1. Latent reasoning contribution unclear at small budgets
In short-budget settings, the method typically switches to explicit mode immediately and does not enter latent reasoning due to the dwell window. Yet it still outperforms CoT, suggesting the improvements largely come from early confidence-based stopping and < think >/< /think > embedding conditioning rather than latent reasoning itself. Ablation isolating these effects would clarify the true source of gains.
2. Scope limited to math-focused tasks
Experiments are concentrated on math reasoning, and it remains unclear whether the method benefits broader domains (e.g., multi-hop QA, code, commonsense reasoning). Reasoning dynamics can differ significantly across tasks.
3. Large-model scaling and SoftThinking baseline discrepancy
The original SoftThinking results indicate improvements over CoT primarily for ≥32B models. Here, the reproduced SoftThink baseline is weaker than CoT on ≤8B models. Given that SwiReasoning is training-free, extending evaluation to ≥32B models would help verify baseline fidelity and clarify whether the proposed method scales beyond the small-model setting.
Heuristic hyperparameters + no runtime reporting
4.Heuristic switching and lack of runtime analysis
The switching logic, dwell window, and mixing schedule are manually designed. While intuitive, their tuning behavior across models and tasks remains unclear, and no wall-clock or FLOP analysis is provided, making deployment efficiency uncertain.

**Questions:**

1 In small budget settings, the model often does not actually enter latent mode due to the dwell window, yet achieves strong gains. Can you provide ablations where latent reasoning is disabled but the early-exit and <think> conditioning remain, to isolate the effect of latent reasoning itself?

2 Token efficiency does not necessarily imply inference-time efficiency. Can you report wall-clock latency or FLOPs vs CoT and SoftThinking to demonstrate practical gains?

3 The SoftThinking baseline underperforms CoT in your ≤8B experiments, whereas the original paper shows gains mainly at ≥32B scale. Can you confirm your implementation matches the original, and provide results on a ≥32B model to ensure fairness?

4 Could you clarify whether your latent update uses the full vocabulary distribution or a top-k truncation (as in SoftThinking)?

5 Could you provide a systematic component-wise ablation (turning each mechanism on/off) to quantify the marginal contribution of:  (i) dwell window (incl. 0) and asymmetry, (ii) < think >/< /think > mixing, (iii) switch-count cap/termination triggers。

---

> ### Author Response · Authors · 2025-11-21
> **Responses to Reviewer hrDS [1/6]**
>
> Dear Reviewer hrDS,
>
> We are glad you found the training-free nature of our work practically useful. Thank you for recognizing the consistency of gains under token budgets and the supportive ablation studies. We appreciate your constructive comments to help strengthen the paper. Your concerns are addressed as follows.
>
> ---
>
> **Q1 & W1. Latent mode behaviors at small budgets and ablations**
>
> Thank you for your perspective.
>
> * Our results demonstrate improvements across various length settings. The latent mode behaviors at very small budgets are one of the design choices. Specifically, we provide users with the flexibility to control token budgets, and small budgets are intended primarily for easy problems. In such scenarios, there are not many diverse reasoning paths to explore, and therefore, the models are supposed to be confident and mainly do CoT exactly. For normal reasoning with bigger budgets, Tab. 1 demonstrates the value of latent.
>
> * We thank you for your suggestions on isolating the effect of the latent. As requested, we have added **a new ablation study in Appendix C. 8**.
>
>   * We used Qwen3-1.7B and compared the full SwiReasoning against a variant where latent reasoning is disabled (SwiR w/o latent). In the "w/o latent" setting, the model is forced to remain in explicit mode, but retains the same switch count controller (functioning as a step-limit controller for inserting the proposed convergence and termination triggers). We evaluated both methods under small token budgets.
>
> | Method / Accuracy (%)     | GSM8K       | MATH 500     | GPQA Diamond | AIME 2024    | AIME 2025    | Average      |
> |---------------|------------|-------------|-------------|-------------|-------------|-------------|
> | SwiR w/o latent | 85.44      | 68.60       | 26.26       | 2.50        | 0.42        | 36.64 |
> | SwiR          | **86.80** | **72.40**  | **29.80**  | **5.83** | **4.58**  | **39.88**  |
> | $\Delta$ |  +1.36  | +3.80 | +3.54 | +3.33  | +4.16 | +3.24 |
>
> | Method / Generation Length | GSM8K | MATH 500 | GPQA Diamond | AIME 2024 | AIME 2025 | Average      |
> |---------------|------|---------|-------------|----------|----------|-------------|
> | SwiR w/o latent | **816**  | 1207    | 1032        | 835      | 791      | 936 |
> | SwiR          | **816**  | **1189**    | **933**         | **812**      | **787**      | **907** |
> | $\Delta$ |   |  |  |   |  | -3.10% |
>
>   * Tab. 8 in the updated version shows that disabling latent leads to a consistent degradation in performance across all benchmarks. Under lower or the same token usage, the full SwiReasoning achieves an average accuracy improvement of +3.24% over the ablation. On challenging benchmarks like AIME2025, the contribution of latent is particularly pronounced (+4.16%). These results show that **the latent itself provides gains that cannot be attributed to token controller mechanisms alone**.
>
> ---

---

> ### Author Response · Authors · 2025-11-21
> **Responses to Reviewer hrDS [2/6]**
>
> **Q2 & W4. Wall-clock latency/TFLOPs & Heuristic hyperparameters**
>
> | Latency (s) | Acc=95% | Acc=90% | Acc=80% |
> |--------------|-----------------|-----------------------------|-------------------------------|
> | CoT          |  99.9 | 88.24 | 71.54 |
> | CoT (Greedy) | 103.07 | 86.03 |  69.66 |
> | Soft Thinking | 100.27 | 88.35 | 71.25 |
> | SwiR (Ours)     | **84.89**  | **64.82** | **32.99** |
> | $\Delta$ | 1.18x faster | 1.36x faster | 2.17x faster |
>
> | TFLOPs | Acc=95% | Acc=90% | Acc=80% |
> |--------------|-----------------|-----------------------------|-------------------------------|
> | CoT          | 78.69  | 70.13  | 57.04 |
> | CoT (Greedy)    | 82.40 | 69.25 | 55.75 |
> | Soft Thinking  | 79.08  | 70.24 | 57.03 |
> | SwiR (Ours)     | **68.30** | **51.72** | **26.22** |
> | $\Delta$ | 1.15x fewer | 1.36x fewer | 2.18x fewer |
>
> * As you suggested, to provide efficiency measurements in more metrics, we have added new **wall-clock latency** and **TFLOPs** experiments in **Appendix C.4**. **Fig. 7** in the updated version demonstrates that SwiReasoning consistently reaches a given accuracy with lower latency and fewer TFLOPs, and the advantage grows as the budget becomes tighter. Specifically, at 90% pass@1 accuracy, SwiReasoning uses 1.36× fewer TFLOPs and is 1.36× faster in wall-clock time than CoT. At 80% accuracy, it uses 2.18× fewer TFLOPs and is 2.17× faster. These results confirm that SwiReasoning improves efficiency at fixed accuracies, especially under small budgets.
>
> * We understand the concern regarding “heuristic”. We wish to clarify that the “heuristic” is largely internal to the framework’s design, not user-facing. Most components are **fixed** across all benchmarks and models, requiring no per-task tuning. For instance, the asymmetric dwell window size of 512, signal mixing coefficient $\beta_0$=0.7, and switch logic described in Sec. 3.3. Further details are also summarized in Appendix B.3.
> Our design philosophy is to provide users with an appropriate level of **flexibility to trade off between latency and accuracy by exposing minimal hyperparameters**, which we believe are necessary for a practical, training-free reasoning framework. In practice, users are provided with the flexibility to adjust $\alpha_0$ based on task difficulty, and optionally, for further efficiency, $C_{max}$ based on budget constraints. All other designs and hyperparameters are fixed inside the framework. This flexibility allows users to adapt SwiReasoning to different task difficulties and efficiency requirements. Tab. 2 and 3 in the paper show that adjustable hyperparameters are not overly sensitive within a broad plateau, and the chosen values of the fixed ones perform well across various scenarios.
>
> ---

---

> ### Author Response · Authors · 2025-11-21
> **Responses to Reviewer hrDS [3/6]**
>
> **W2. Experiments on broader domains: code, multi-hop QA, and commonsense reasoning**
>
> Thank you for your suggestion.
>
> * We’d like to clarify that our original experiments **already include task types other than mathematics**. Specifically, GPQA Diamond is a STEM reasoning dataset that includes problems in physics, chemistry, biology, and computer science, in addition to mathematics.
>
> * We agree that demonstrating generalizability on more task types is important. As you requested, we conducted **four new experiments** (detailed in **Appendix C.3**). In addition to the math and STEM domains, we also evaluated SwiReasoning in
> **coding**, **multi-hop QA**, and **commonsense reasoning**. We use Qwen3-8B and report pass@1 accuracy on HumanEval [1] and LeetCode-Contest [2] for coding, 2WikiMultihopQA [3] set from LongBench [4] for multi-hop QA, and CommonsenseQA [5] for commonsense reasoning. As shown in Tab. 6 of the updated version, under no token budget constraint, SwiReasoning improves accuracy by +3.40% on average over standard CoT (the reported average is the simple mean of accuracy over the full LeetCode-Contest and accuracies over three other benchmarks).
> | Method         | HumanEval | LeetCode-Easy | LeetCode-Medium | LeetCode-Hard | 2WikiMultihopQA | CommonsenseQA | Average    |
> |----------------|-----------|-----------------------|-------------------------|-----------------------|-----------------|---------------|-----------|
> | CoT            | 92.68     | 57.78                 | 68.13                   | 43.18                 | 79.00           | 83.95         | 78.77   |
> | CoT (Greedy)   | 93.90     | 64.44                 | 58.24                   | 47.73                 | 79.50           | 83.95         | 78.64  |
> | Soft Thinking  | 92.07     | 55.56                 | 61.54                   | 38.64                 | 79.00           | 83.70         | 77.30  |
> | **SwiR (Ours)**| **95.73** | **64.44**  | **69.23**  | **61.36**  | **81.50**  | **85.34**  | **82.17**  |
> | $\Delta$ | +3.05 | +6.66 | +1.10 | +18.18 | +2.50 | +1.39 | +3.40 |
>
>
>   * On **coding** tasks, the largest gains (+18.18%) are observed on the hard-level subset, indicating that SwiReasoning is most helpful for problems that require stronger reasoning capabilities.
>   * On **multi-hop QA** tasks, which require retrieving and connecting multiple disparate facts, SwiReasoning outperforms CoT by +2.50%. This suggests that the exploration capability of latent reasoning is effective in navigating complex reasoning paths.
>   * On **commonsense reasoning** tasks, SwiReasoning surpasses CoT by +1.39%. This demonstrates the robustness of SwiReasoning in general knowledge scenarios.
>
> Overall, these results confirm that SwiReasoning **generalizes to broader domains with consistent accuracy gains**.
>
> ---

---

> ### Author Response · Authors · 2025-11-21
> **Responses to Reviewer hrDS [4/6]**
>
> **Q3 & W3. Experiments with a >= 32B model and the baseline performance**
>
> | Method         | GSM8K        | MATH 500     | GPQA Diamond | AIME 2024    | AIME 2025    | Average        |
> |---------------|-------------:|-------------:|-------------:|-------------:|-------------:|---------------:|
> | CoT           | 95.83        | 97.40        | 66.16        | 80.42        | 72.08        | 82.38  |
> | CoT (Greedy)  | 95.91        | 97.20        | 69.70        | 80.00        | 73.33        | 83.23  |
> | Soft Thinking | 95.75        | 97.40        | 67.17        | 74.58        | 66.25        | 80.23  |
> | SwiR (Ours)   | **96.21** | **98.40** | **70.20** | **82.92** | **73.75** | **84.30** |
> | $\Delta$ | +0.38 | +1.00 | +4.04 | +2.50 | +1.67 | +1.92 |
>
> * **Regarding the large-scale Qwen3-32B model**: We agree that demonstrating scalability on larger models could strengthen our empirical results. As you requested, we conducted new experiments on Qwen3-32B in Appendix C.2. As shown in Table 5 of the updated version, SwiReasoning consistently improves performance, achieving an average gain of +1.92% over standard CoT under no token budget constraint. The gains are most notable on more difficult benchmarks such as GPQA Diamond (+4.04%). These results confirm that SwiReasoning **scales effectively to larger models**.
>
> * **Regarding the baseline performance**: We thank you for noting this. We would like to clarify that we are using the correct implementation, and this performance discrepancy is not observed in our work alone. A concurrent work [6] also reported a **similar observation independently**. For example, in Tab. 1 of their paper, they found that soft thinking underperforms standard CoT with sampling, with an average accuracy decrease of 2.56%- 6.37%.
> We want to highlight that soft thinking is a very inspiring and insightful work. For this performance discrepancy observed by our work and the concurrent work, we believe it may be largely because soft thinking was originally evaluated on relatively older model families. As reasoning LLMs have been evolving fast recently, their baseline CoT capabilities have improved significantly. Overall, we assure the reviewer that the comparisons are **fair**.
>
> ---
>
> **Q4. About the latent update strategy**
>
> Thank you for asking. We use the **full** vocabulary distribution as described in Eq. 1:
>
> $$
> \tilde{e}_{t} = \sum_{v\in V}p_{t}[v]e^{(v)},
> $$
>
> which is a sum over the entire vocabulary $V$, with no top-k/top-p truncation.
>
> ---

---

> > ### Comment · Reviewer_hrDS · 2025-11-21
> >
> > Thanks, the results partially addressed my concern. Can you test your method on other >= 32B model (Deepseek-R1-Distill-Qwen-32B, QwQ32B) and other benchmark (LiveCodeBench, MBPP)?

---

> ### Author Response · Authors · 2025-11-21
> **Responses to Reviewer hrDS [5/6]**
>
> **Q5 & W4. Component-wise ablation**
>
> Thank you for your suggestion. We provide these requested ablations in **Appendix C.9**. To quantify the marginal contribution of each proposed mechanism, we conducted a component-wise ablation study using Qwen3-1.7B on GSM8K and MATH500 benchmarks.
>
> | Method                                     | GSM8K  | Math500 | Average | Chosen |
> |--------------------------------------------|--------|---------|---------|---------|
> | *SwiR* w/ window size 0                    | 89.00% | 89.20%  | 89.10%  | |
> | *SwiR* w/ symmetry window size 2           | 89.31% | 90.00%  | 89.66%  | |
> | *SwiR* w/ symmetry window size 4           | **90.83%** | 89.20%  | 90.02%  | |
> | *SwiR* w/ symmetry window size 8           | 90.60% | 91.20%  | 90.90%  | |
> | *SwiR* w/ symmetry window size 16          | 88.93% | 92.00%  | 90.47%  | |
> | *SwiR* w/ symmetry window size 32          | 90.14% | 91.00%  | 90.57%  | |
> | *SwiR* w/ asymmetry window size 64         | 89.69% | 92.60%  | 91.15%  | |
> | *SwiR* w/ asymmetry window size 128        | 90.45% | 91.00%  | 90.72%  | |
> | *SwiR* w/ asymmetry window size 256        | 89.76% | 90.80%  | 90.28%  | |
> | *SwiR* w/ asymmetry window size = 512     | **90.83%** | **93.00%** | **91.92%** | ✓ |
> | *SwiR* w/ asymmetry window size = 1024     | **90.83%** | 91.20%  | 91.01%  | |
> | *SwiR* w/o mixing                          | 90.44% | 92.20%  | 91.32%  | |
>
>  * **(i) Dwell window (incl. 0) and asymmetry**. We compared the default asymmetric window ($W_{E \rightarrow L}=512, W_{L \rightarrow E}=0$) against variations with no window ($W=0$) and symmetric windows of varying sizes. Results in Tab.9 of the updated version reveal that
>
>    * **Necessity of a window**: Removing the window constraint completely leads to a significant performance drop (Avg: 89.10% vs. 91.92%), due to rapid oscillations between modes preventing coherent reasoning chains from forming.
>    * **Benefit of asymmetry**: The default asymmetric setting outperforms all symmetric configurations. While symmetric windows (*e.g.*, size 8) improve over no window, they fail to match the peak performance of the asymmetric design. This validates our hypothesis that explicit reasoning requires a consolidation period ($W_{E \rightarrow L}$), while latent reasoning benefits from immediate exit upon confidence recovery ($W_{L \rightarrow E}=0$) to reduce the risks of introducing spurious signals that may mislead the model.
>
> * **(ii) Signal mixing**. Tab.9 also indicates that disabling the signal mixing results in a consistent accuracy decrease (Avg: 91.32% vs. 91.92%). This suggests that the signal mixing helps the model better orient its internal state during mode transitions and contributes positively to the final accuracy.

---

> ### Author Response · Authors · 2025-11-21
> **Responses to Reviewer hrDS [6/6]**
>
> | $C_{\max}$        | 4      | 8      | 12     | 16     | 20     | 24     |
> |---------------------|--------|--------|--------|--------|--------|--------|
> | Accuracy            | 72.40% | 81.00% | 85.00% | 86.80% | 87.80% | 87.80% |
> | Generation Length   | 1189   | 2222   | 2914   | 3280   | 3597   | 3862   |
>
> | $C_{\max}$        | 28     | 32     | 36     | 40     | 44     | $\infty$ |
> |---------------------|--------|--------|--------|--------|--------|-----------|
> | Accuracy            | 89.00% | 89.20% | 89.80% | 90.60% | 91.80% | **93.00%** |
> | Generation Length   | 4068   | 4186   | 4274   | 4318   | 4602   | 4924      |
>
> * **(iii) switch-count cap/termination triggers.** We analyzed the effect of the switch count cap ($C_{max}$) on accuracy and generation length using Qwen3-1.7B on MATH500 in Tab. 10 of the updated version. The results indicate that
>    * **Efficiency control**: As $C_{max}$ decreases (from $\infty$ down to 4), the generation length drops significantly (from 4924 to 1189), demonstrating the controller's effectiveness in curbing overthinking. This provides users with sufficient capabilities to control the intensity of thinking flexibly.
>    * **Accuracy trade-off**: While extremely tight caps (*e.g.*, 4 or 8) limit accuracy, moderate caps (*e.g.*, 20-32) achieve performance competitive with or close to the unlimited setting ($C_{max}=\infty$) but with much lower token consumption. This confirms that the controller effectively makes use of partial reasoning trajectories to deliver reasonable answers.
>
> ---
>
> Thank you again for your thoughtful suggestions, which have motivated us to conduct additional experiments that have strengthened the paper. We’d appreciate your re-evaluation based on the updated version. Please do not hesitate to let us know if you have any further comments. We will be glad to discuss them.
>
> ---
>
> **References**
> [1] “Evaluating Large Language Models Trained on Code” by OpenAI, 2021.
> [2] “DeepSeek-Coder: When the Large Language Model Meets Programming - The Rise of Code Intelligence” by DeepSeek, 2024.
> [3] “Constructing A Multi-hop QA Dataset for Comprehensive Evaluation of Reasoning Steps” by Ho et al. COLING 2020.
> [4] “Longbench: A bilingual, multitask benchmark for long context understanding” by Bai et al. ACL 2024
> [5] “Commonsenseqa: A question answering challenge targeting commonsense knowledge” by Talmor et al. NAACL 2019.
> [6] LLMs are Single-threaded Reasoners: Demystifying the Working Mechanism of Soft Thinking. ArXiv’25.

---

> ### Author Response · Authors · 2025-11-26
> **Responses to Reviewer hrDS [7/8]**
>
> Dear Reviewer hrDS,
>
> We sincerely thank you for your prompt and positive feedback, and we appreciate your follow-up questions on additional experiments with older models and more coding benchmarks. Over the past few days, we have conducted these additional experiments, and are pleased to report more results to address your follow-up concerns as follows.
>
> ---
>
> **Q6. Experiments with older models**
>
> As requested, we further evaluate SwiReasoning with the Deepseek-R1-Distill-Qwen-32B model, which was adopted by Soft Thinking’s original paper, on Math500. The results validate the fairness of baseline comparisons further.
>
> | Method                                      | Math500 |
> |---------------------------------------------|---------|
> | CoT                                         | 95.00%   |
> | CoT (Greedy)                                | 92.60%   |
> | Soft Thinking (threshold=0.01, cold_k=256)  | 94.20%   |
> | Soft Thinking (threshold=0.01, cold_k=512)  | 94.20%   |
> | Soft Thinking (threshold=0.01, cold_k=1024) | 94.20%   |
> | Soft Thinking (threshold=0.05, cold_k=256)  | 94.60%   |
> | Soft Thinking (threshold=0.05, cold_k=512)  | 94.00%   |
> | **Soft Thinking (threshold=0.05, cold_k=1024)** | **95.20%** |
> | Soft Thinking (threshold=0.1, cold_k=256)   | 94.40%   |
> | Soft Thinking (threshold=0.1, cold_k=512)   | 93.40%   |
> | Soft Thinking (threshold=0.1, cold_k=1024)  | 93.40%   |
> | **SwiR (Ours)**                             | **96.20%** |
> | $\Delta$ | +1.20% |
>
> As shown in the table below, SwiReasoning (96.20%) achieves the best performance, surpassing both CoT and the optimal configuration of Soft Thinking. More specifically, we tested all 9 hyperparameter combinations recommended by the Soft Thinking paper as follows,
>
> ```bash
> 1. threshold = 0.01, cold_k=256,
> 2. threshold = 0.01, cold_k=512,
> 3. threshold = 0.05, cold_k=1024,
> 4. threshold = 0.05, cold_k=256,
> 5. threshold = 0.05, cold_k=512,
> 6. threshold = 0.1, cold_k=1024,
> 7. threshold = 0.1, cold_k=256,
> 8. threshold = 0.1, cold_k=512,
> 9. threshold = 0.1, cold_k=1024.
> ```
>
> Among which Soft Thinking achieves its highest performance, 95.20% accuracy, when threshold=0.05 and cold_k=1024. This reproduced performance is closely matched with the number 95.00% reported in the original paper, as shown below, validating that we are using the correct implementation.
>
> | Method                           | Math500 |
> |----------------------------------|---------|
> | CoT (reported)                   | 94.50   |
> | CoT Greedy (reported)            | 93.00   |
> | Soft Thinking (reported)         | 95.00   |
> | Soft Thinking (reproduced max)   | 95.20   |
> | Soft Thinking (reproduced mean)  | 94.18   |
> | **SwiR (Ours)**                  | **96.20** |
> | $\Delta$ | +1.20% |
>
> Taking our existing results on the newer Qwen3-32B model together, these results demonstrate that SwiReasoning **scales effectively** to 32B models with **fair** comparisons to baselines.
>
> ---

---

> > ### Comment · Reviewer_hrDS · 2025-11-26
> >
> > Thanks for the experiment, I raised the score.

---

> > > ### Author Response · Authors · 2025-11-26
> > >
> > > Thank you for your positive feedback and for raising the score. We are glad that our responses addressed your concerns. Your constructive suggestions have been invaluable in strengthening this work!

---

> ### Author Response · Authors · 2025-11-26
> **Responses to Reviewer hrDS [8/8]**
>
> **Q7. Experiments on more coding benchmarks**
>
> As requested, we have extended our experiments on coding beyond HumanEval and LeetCode-Contest to include the MBPP [7] and LiveCodeBench [8] benchmarks in the updated Appendix C.3.
>
> | Method                                       | MBPP  |
> |----------------------------------------------|-------|
> | CoT                                          | 94.16 |
> | CoT (Greedy)                                 | 91.44 |
> | Soft Thinking (threshold=0.01, cold_k=256)   | 93.39 |
> | **Soft Thinking (threshold=0.01, cold_k=512)** | **94.16** |
> | Soft Thinking (threshold=0.01, cold_k=1024)  | 93.77 |
> | Soft Thinking (threshold=0.05, cold_k=256)   | 93.77 |
> | Soft Thinking (threshold=0.05, cold_k=512)   | 93.00 |
> | Soft Thinking (threshold=0.05, cold_k=1024)  | 93.39 |
> | Soft Thinking (threshold=0.1, cold_k=256)    | 93.38 |
> | Soft Thinking (threshold=0.1, cold_k=512)    | 93.00 |
> | Soft Thinking (threshold=0.1, cold_k=1024)   | 93.39 |
> | **SwiR (Ours)**                              | **95.33** |
> | $\Delta$ | +1.17% |
>
> | Method                                        | LiveCodeBench |
> |-----------------------------------------------|----------------|
> | CoT                                           | 62.01          |
> | CoT (Greedy)                                  | 50.18          |
> | Soft Thinking (threshold=0.01, cold_k=256)    | 56.63          |
> | **Soft Thinking (threshold=0.01, cold_k=512)** | **56.99**      |
> | Soft Thinking (threshold=0.01, cold_k=1024)   | 55.56          |
> | Soft Thinking (threshold=0.05, cold_k=256)    | 56.27          |
> | **Soft Thinking (threshold=0.05, cold_k=512)** | **56.99**      |
> | Soft Thinking (threshold=0.05, cold_k=1024)   | 56.27          |
> | Soft Thinking (threshold=0.1, cold_k=256)     | 56.27          |
> | **Soft Thinking (threshold=0.1, cold_k=512)**  | **56.99**      |
> | Soft Thinking (threshold=0.1, cold_k=1024)    | 56.27          |
> | **SwiR (Ours)**                               | **63.44**      |
> | $\Delta$ | +1.43% |
>
> The results above show that the SwiReasoning achieves substantial gains on MBPP (+1.17%) and LiveCodeBench (+1.43%) benchmarks as well.
>
> Integrating these new tasks, SwiReasoning demonstrates **consistent improvements** across Coding, Multi-hop QA, and Commonsense Reasoning. The updated table below shows an average accuracy improvement of +2.70% over standard CoT across various domains other than Mathematics and STEM.
>
> | Method         | HumanEval | LeetCode-Easy | LeetCode-Medium | LeetCode-Hard | MBPP | LiveCodeBench | 2WikiMultihopQA | CommonsenseQA | Average    |
> |----------------|-----------|-----------------------|-------------------------|-----------------------|-----------------|-----------------|-----------------|---------------|-----------|
> | CoT            | 92.68%     | 57.78%                 | 68.13%                   | 43.18%                 | 94.16% | 62.01% | 79.00%           | 83.95%         | 78.54%   |
> | CoT (Greedy)   | 93.90%     | 64.44%                 | 58.24%                   | 47.73%              | 91.44% |   50.18% | 79.50%           | 83.95%         | 76.03%  |
> | Soft Thinking  | 92.07%     | 55.56%                | 61.54%                   | 38.64%               | 94.16% |  56.99% | 79.00%           | 83.70%         | 76.73%  |
> | **SwiR (Ours)**| **95.73%** | **64.44%**  | **69.23%**  | **61.36%**  | **95.33%** | **63.44%** | **81.50%**  | **85.34%**  | **81.24%**  |
> | $\Delta$ | +3.05% | +6.66% | +1.10% | +18.18% | +1.17% | +1.43% | +2.50% | +1.39% | +2.70% |
>
>
> ---
>
> Once again, we thank you for your constructive feedback, which has helped strengthen this work. We'd genuinely appreciate your re-evaluation/further comments!
>
> ---
>
> **References (Cont.)**
> [7] “Program Synthesis with Large Language Models” by Austin et al. ArXiv’21.
> [8] “Livecodebench: Holistic and contamination free evaluation of large language models for code” by Jain et al. ArXiv’ 24.

---

### Author Response · Authors · 2025-12-03
**Thank You and Summary for Submission 20430**

Dear Reviewers, ACs, SACs, and PCs,

Thank you for your time and effort in ensuring a high-quality review process for ICLR. We respectfully provide a summary for the ACs’ reference to document what happened and how each concern was addressed.
|Reviewer|hrDS|dZdK|eVQp|BFnL|JcUf|
|---|---|---|---|---|---|
|**Final Rating***|**6**|?|**8**|?|**6**|
|Initial Rating|2|4|6|4|6|
|Response|Concerns addressed|Not provided|Concerns addressed|Not provided| Concerns addressed|
|Link|[Response1](https://openreview.net/forum?id=t33kMzEAg8&noteId=aAGbI0Hlri) & [2](https://openreview.net/forum?id=t33kMzEAg8&noteId=Kj6SLKLTst) (both before the incident)||[Response](https://openreview.net/forum?id=t33kMzEAg8&noteId=7z4TS2GSt3)||[Response](https://openreview.net/forum?id=t33kMzEAg8&noteId=QNQq8VDd2t)|

$^{*}$ Evidenced by "Revisions" buttons in reviews.

Our rebuttals and extensive new experiments received acknowledgment from multiple reviewers and secured **updated ratings 86644** by Nov. 27. Three reviewers stated that our responses resolved their concerns, and two raised ratings. Due to the early-terminated discussion, we haven’t heard back from the two reviewers with initial rating 4.

---
1. We sincerely appreciate positive assessments from reviewers, who **highlighted the novelty addressing prior limits (eVQp, dZdK), intuitive & effective design (hrDS, dZdK, eVQp, BFnL), comprehensive experiments (dZdK, eVQp, JcUf), consistent gains (hrDS, eVQp), practical value (JcUf, hrDS), and supportive ablations (JcUf, dZdK, hrDS)**.
---
2. **We fully addressed Reviewer hrDS, eVQp, and JcUf’s concerns**
   * They requested: 1) adding models of >= 32B, 2) adding more tasks (coding, multi-hop reasoning, and commonsense QA), 3) wall-clock latency & TFLOPs measurements, 4) histograms of switch dynamics, 5) challenging case analyses, 6) baseline discrepancy investigations, and 7) extra ablations.
   * **We thoroughly conducted all of these as requested**, validating this work 1) scales effectively to larger models (Qwen3-32B, Deepseek-R1-Distill-Qwen-32B), 2) generalizes to various domains (6 additional benchmarks on coding, multi-hop reasoning, and commonsense QA) with +2.70% accuracy gains on average in training-free, 3) achieves significantly improved wall-clock latency & TFLOPs, 4) enables dynamic adaptation to problem difficulty, 5) analyzes trade-offs in latent mode, 6) ensures fair comparisons, 7) substantiates component contributions.
   * **All three reviewers explicitly recognized these updates**. After we addressed partial 1), partial 2), 3), 6), 7), Reviewer **hrDS** confirmed “Thanks, the results partially addressed my concern,” and raised the rating to 4 on Nov. 21 (before the incident). After we fully addressed 1), 2), Reviewer **hrDS** followed up with “Thanks for the experiment, I raised the score,” and finalized the rating as 6 on Nov. 26 (before the incident). After we addressed 4), 5), 7), Reviewer **eVQp** endorsed “These results further make the proposed approach strong. I will increase the score,” and finalized the rating as 8 on Nov. 27. After we addressed 2) and clarified our design principles and gains, Reviewer **JcUf** affirmed “The authors’ clarifications resolved my concerns” and kept the positive rating 6 on Nov. 27.
---
3. **We believe we fully addressed Reviewer dZdK and BFnL’s concerns**
   * They requested: 1) adding larger models, 2) adding more tasks, 3) adding other efficiency metrics, and 4) adding evidence for latent exploration.
   * **We thoroughly conducted all of these as requested**. Items 1), 2), 3) are a subset of 1), 2), 3) that we had addressed for other reviewers. For 4), we provided clear evidence from entropy and semantic diversity. Reviewer dZdK asked for clarifications on design choices and inaccurate BibTeX entries. We clarified our design principles and checked all entries to fix two of them. To Reviewer BFnL’s question on gains, we provided more evidence on latency and outlined substantial gains achieved in training-free. To Reviewer BFnL’s confusion about Tab. 8, we clarified our twofold contributions based on budget settings. To Reviewer BFnL’s question on why it works and the interpretability, we elaborated the key rationales described in Sec. 3.3 and explained the strategy for analyzing interpretability.
   * Because the discussion stopped early,  we haven’t heard back. However, the new experiments and clarifications that address their concerns are already in the revised version, and **substantially overlap** with the requests from other reviewers who stated that **their concerns were resolved** and, in two cases, raised their scores.
---
**All requested modifications have been incorporated faithfully into the updated paper** (highlighted in red). Given the context, we respectfully request that ACs take our points into consideration while making the final decision. Thank you again for your valuable time and thoughtful consideration.

Best regards,
Authors of Submission 20430

---

### Meta-Review · Area_Chair_gfUR · 2026-01-05

**Summary:**

This paper was reviewed by five experts in the field. Three of the five reviewers acknowledged that their concerns were addressed and raised their scores. The two remaining reviewers (both initially scoring 4) did not respond before the discussion closed, but their concerns closely mirrored those already resolved. Thus, the AC believes that their scores would likely improve to at least 6. The AC has checked the paper and the rebuttal and found that the concerns have been mostly addressed. However, the AC strongly encourages the authors to add the results of larger models (Qwen3-32B) and additional tasks into the main paper.

Given the strong empirical validation and consistent gains across models and tasks, the AC recommends accept for this paper.

**Reviewer Concerns:**

1. Performance on large models and more tasks: The authors have included Qwen3-32B, Deepseek-R1-Distill-Qwen-32B, and evaluated the generalization ability on 6 additional benchmarks.

2. The actual latency and FLOPS: The authors have provided a detailed efficiency analysis in the rebuttal.

3. Analysis of the switch dynamics: The authors have provided in the rebuttal and verified that the proposed method enables dynamic adaptation to the difficulties of different problems.

4. Baseline discrepancy: The authors presented the experiments with a >= 32B model and the baseline performance, and clarified that they have used the correct implementation to ensure a fair comparison.

5. Other clarity issues and ablation studies: The authors have provided the necessary revisions to evidence on latency and outlined substantial gains achieved in training-free, and elaborated rationales described in Sec. 3.3 and explained the strategy for analyzing interpretability.

**Reviewer Scores:**

The initial scores were 2, 4, 6, 4, and 6. Three reviewers have since responded to the authors’ rebuttal, and two of them were willing to increase the scores, resulting in updated scores of 6, 4, 8, 4, and 6.

The AC has checked the discussion and confirms that the concerns raised by the three responding reviewers have been addressed. As for the two remaining reviewers, both of whom originally gave a score of 4, their concerns have largely been resolved, and one or both may raise their scores to 6.

---

### Decision · Program_Chairs · 2026-01-26

Accept (Poster)